# Automatic pollen recognition with the Rapid-E particle counter: the first-level procedure, experience and next steps

Ingrida Šaulienė[1], Laura Šukienė[1], Gintautas Daunys[1], Gediminas Valiulis[1], Lukas Vaitkevičius[1], Predrag Matavulj[2], Sanja Brdar[2], Marko Panic[2], Branko Sikoparija[2], Bernard Clot[3], Benoît Crouzy[3] Mikhail Sofiev[1,4]

[1]Siauliai University, Šiauliai, 76352 Lithuania
[2] BioSense Institute - Research Institute for Information Technologies in Biosystems, University of Novi Sad, Novi Sad, 21000, Serbia
[3]Federal Office of Meteorology and Climatology MeteoSwiss, Payerne, CH-1530, Switzerland
[4]Finnish Meteorological Institute, Helsinki, 00560, Finland

*Corespondence to*: Ingrida Šaulienė (ingrida.sauliene@su.lt)

**Abstract.** Pollen-induced allergy is among the most-prevalent non-contagious diseases, with about a quarter of European population sensitive to various atmospheric bioaerosols. In most European countries, pollen information is based on a weekly-cycle Hirst-type pollen trap method. This method is labour-intensive, requires narrow specialization abilities and substantial time, so that the pollen data are always delayed, subject to sampling- and counting-related uncertainties. Emerging new approaches to automatic pollen monitoring can, in principle, allow for real-time availability of the data with no human involvement.

The goal of the current paper is to evaluate the capabilities of the new Plair Rapid-E pollen monitor and to construct the first-level pollen recognition algorithm. The evaluation was performed for three devices located in Lithuania, Serbia and Switzerland, with independent calibration data and classification algorithms. The Rapid-E output data include multi-angle scattering images and the fluorescence spectra recorded at several times for each particle reaching the device. Both modalities of the Rapid-E output were treated with artificial neural networks (ANN) and the results were combined to obtain the pollen type. For the first classification experiment, the monitor was challenged with a large variety of pollen types and the quality of many-to-many classification was evaluated. It was shown that in this case, both scattering- and fluorescence- based recognition algorithms fall short of acceptable quality. The combinations of these algorithms performed better exceeding 80% accuracy for 5 out of 11 species. Fluorescence spectra showed similarities among different species ending up with three well-resolved groups: (*Alnus*, *Corylus*, *Betula* and *Quercus*), (*Salix* and *Populus*), and (*Festuca*, *Artemisia*, *Juniperus*). Within these groups, pollen is practically non-distinguishable for the first-level recognition procedure. Construction of multi-steps algorithms with sequential discrimination of pollen inside each group seems to be one of possible ways forwards. In order to connect the classification experiment to existing technology, a short comparison with the Hirst measurements is presented and an issue of the false-positive pollen detections by Rapid-E is discussed.

Key words: pollen observations, real-time monitoring, artificial neural networks (ANN), scattering, fluorescence


## 1 Introduction

Pollen of many wind-pollinated plants has specific proteins that cause human allergy (Valenta et al., 1992;
Bousquet et al., 2006; Radauer and Breiteneder, 2006; Bousquet et al., 2015; Choual et al. 2018),
particularly affecting children (Skoner 2001; Höst et al., 2003; Douladiris et al., 2018). Stress due to
contact with the pollen-contained allergen can cause an allergic reaction or exacerbate some related
diseases (Leynaert et al., 2000; Devillier et al., 2017; Poethko-Müller et al., 2018). Allergy impairs the
quality of life of about 30% of the world population (Akdis et al., 2015). In most of European countries,
national organizations of various kinds provide information about pollen concentration in the air, publish
pollen prognosis and issue warnings. The bulk of such efforts are based on retrospective pollen
observations and climatological pollen calendars. Most observers use Hirst-type pollen traps and samples
are identified by a microscopic analysis (Galán et al., 2014; Buters et al., 2018). This method is labour-
intensive, tedious, requires narrow specialization abilities and incorporates significant uncertainties – e.g.
(Oteros et al., 2017). In addition, due to the manual treatment of the collected samples and weekly cycle
of the trap the data are always delayed from a few days up to a few weeks. However, timely data about
pollen concentration in the air are also needed for improving the accuracy of tools for personalized
medicine (for example, PASYFO app, http://www.pasyfo.lt, POLLEN app, http://www.polleninfo.org,
NORKKO forecast and app http://www.norkko.fi, etc.) (Bousquet et al., 2017; Horgan and Pazzagli,
2017; Pereira et al., 2018, Tabatabaian and Casale 2018). It can be also used for informing people about
current pollen concentration in the air. Finally, real-time data are needed for short-term pollen forecasts
with statistical and atmospheric dispersion models (Sofiev et al, 2013, 2015, 2017, Prank et al., 2016;
Ritenberga et al., 2016; Zink et al., 2017).

As the approach to information and personal responsibility for health is changing, it becomes necessary
to develop new methods enabling the information on airborne pollen available in real time. The first
attempts to obtain automated information are related to image recognition technologies (Bennett, 1990).
Their development was accompanied by the formation of more potential possibilities (Ronneberger et al.,
2002; Landsmeer, 2009). Currently, two types of technologies seem to be the most-suitable for taxon-
level classification of pollen: based on image recognition and laser-fluorescence (or their combinations).
Image-based technologies are used in detectors, such as BAA500 (Hund Wetzlar,
https://www.hund.de/en/); the laser fluorescence-based approach is implemented in PA-300 and Rapid-E
(Plair, http://www.Plair.ch), whereas the new Poleno device (Swissens, https://swisens.ch/) aims at
integration of both features. The Hund- and Plair- manufactured devices were used in limited-scale
scientific studies: Oteros et al. (2015) for BAA500 and Crouzy et al. (2016) for PA-300 and showed
promising results. However, the large-scale evaluation and calibration suitable for European-scale
applications are yet to be concluded (Oteros et al., 2015, Crouzy et al., 2016).

The goal of the current paper is to evaluate the capabilities of the new Plair Rapid-E pollen monitor and
to construct and evaluate the first-level pollen recognition algorithm using the Rapid-E data. The
experiment was performed in Siauliai (University of Siauliai and Finnish Meteorological Institute), Novi
Sad (BioSense Institute of University of Novi Sad) and Payerne (MeteoSwiss) with three newly acquired
Rapid-E devices. Below, we present in details the procedure followed in Siauliai and resulting extensive
assessment of the device performance. The best classification result is then compared with the best first-
level classification of same or similar pollen morphotypes made independently in Novi Sad and Payerne.



Finally, outputs of MeteoSwiss classifier in a monitoring setup are presented and compared with the Hirst
data.

## 2 Methods

### 2.1. Description of the measurement instrument

The new Rapid-E instrument designed and produced by Plair S.A is the successor of the first-generation
particle analyser PA-300 used by Crouzy et al. (2016). It is a particle counter, i.e. analyses all particles
coming to its inlet one-by-one. Operation of the instrument is based on two physical principles: scattering
of near-UV laser beam and deep-UV laser-induced fluorescence (Kiselev et al., 2011; 2013). Multi-angle
scattering is used for determination of the particle's morphology, such as size and shape. The fluorescent
light is analysed for its spectrum and lifetime. The instrument constantly takes in the ambient air through
the air inlet on the top of its panel. Sample air flow is up to 2.8 litres per minute with the counting rate of
up to 4500 particle detections per minute, i.e. the theoretical saturation level is $1.6 \ 10^6$ particles m$^{-3}$. Since
the smallest observable particle is 0.5μm in diameter, this range is sufficient for practically all ambient
conditions. The sampled air enters the nozzle, which creates a laminar flow in the measurement zone.
Particles interact with 400 nm laser light source and the scattered light is captured by twenty-four time-
resolving detectors distributed at different angles. The information on chemical properties of the particles
is obtained by a powerful deep-UV laser (320 nm) source that induces fluorescence. Its spectrum (32
measuring channels within spectral range of 350-800 nm, 8 sequential acquisitions with 500 ns retention)
and lifetime (4 particular bands) are recorded and used for the particle identification (Figure 1).
Rapid-E has an embedded mechanism for collecting the particles, which passed through the registration
chamber, onto sticky slides for the follow-up microscopic analysis.
The device has several modes of operations. Since the deep-UV laser has a limited resource, the 400 nm
scattering image is used for prior estimation of the particle morphology and deciding if it can be pollen.
In the Pollen mode, the device ignites the deep-UV laser only for 5-100 micrometer particle size range.
Another mode allows detecting particles in the range of 0.5-100 micrometers for spores, particulate matter
and bacteria identification. However, the expected lifetime of the deep-UV laser is much shorter in this
mode, especially in polluted atmosphere.

### 2.2 The scheme of the experiment

#### 2.2.1 Siauliai

The experiment in Siauliai was carried out with 14 pollen morphotypes, the tested amounts of which are
given in Table 1. Three genera (*Salix, Acer, Pinus*) were represented by two plant types. All 14 plants are
naturally widespread in Lithuania and their airborne pollen is abundantly recorded annually (Šaulienė et
al., 2016). These particles were provided to the device one set after another splitting the recordings to
training and test subsets. After that, the pollen recognition algorithm was calibrated using training subsets



and challenged with pollen from the test subsets with no stratification or a-priori other information provided. Each-to-each confusion table was the main quality measure.

Pollen was taken from the plant inflorescences collected during the vegetation period in April-August of 2018 during the days with intense pollen release. The collected material was put in air-permeable paper bags and dried at a temperature of 40°C until the maximum release of pollen from the inflorescences. Vibratory Sieve Shaker ANALYSETTE 3 PRO was used for gentle shaking the pollen grains out of the inflorescences. The extracted pollen was stored in Petri dishes at +4°C.

Each experiment was repeated twice and consisted of up to eight sample tests, each using approximately 5 mg of pollen. The experiments were carried out in laboratory conditions with self-designed manual exposure method (Figure 2). In order to isolate the environment of the experiment from the ambient particles, a plastic (PET) bottle was fit tightly to the Rapid-E inlet. One of the bottle walls was cut open and two holes of ~15 cm$^2$ were covered with a household air filter. The filter fabric was tested to hold

~99% of particles larger than 1 micrometre in diameter without any noticeable disturbance of the air inflow into the device.
The pollen was injected into the upper part of the bottle by inserting the pipette tip with the pollen sample into the narrow cut in the bottle and then gently blowing the air through the pipette. With the sampling rate up to 2.8 litres per minute, Rapid-E was collecting the pollen grains from the bottle within a few tens

of seconds. This simple scheme enabled reducing the environmental sample contamination by up to 5 times compare to the unfiltered air in the lab. Each new experiment used new bottle and the nozzle of the instrument was cleaned, thus ensuring the removal of previously blown pollen.
Quality and level of contamination of the samples was manually controlled by using the sticky slides. The presence of non-pollen particles (debris from the remnants of inflorescences etc) was verified to be

substantially less than 1% by the visual inspection of a subset of the calibration events. Abundance of pollen aggregates (several pollens stuck together) was also low but their reliable identification by microscopic analysis was more difficult because of busy slides. The calibration was performed in the Pollen mode, which excluded particles smaller than 5 μm of optical size.

### 2.2.2 Novi Sad

The scheme of the experiment was practically identical in Siauliai and Novi Sad. The pollen exposure was conducted on the roof by fitting the PET bottle to the sampling pipe after removing the Sigma2 inlet. Manual microscopic analysis of sticky slides was used to confirm the quality of samples and absence of non-pollen debris and pollen agglomerates. The device was also in Pollen mode, i.e. it filtered out particles smaller than 5 μm of optical size.

Classification was tested for same or similar pollen morphotypes accounting for the availability of the fresh material during the study season. In particular, *Juniperus* was replaced by *Taxus* and *Festuca* was replaced by *Cynodon* and *Dactilis* aiming to assess the degree of discrimination between different grass genera. Similarly, *Picea* and *Cedrus* pollen were used for assessing differences between the same pollen morphotype. Only *Acer negundo* was analysed as it is the only *Acer* pollen that is recorded regularly in

Serbia. *Fraxinus* (including both *F. excelsior* and *F. ornus*) was added to the test as it is commonly recorded throughout spring season.





### 2.2.3 Payerne

For practical reasons, pollen calibrations in Payerne were performed by directly blowing the material into the Sigma 2 inlet, without protection from contamination. The details of the procedure are described in 160 (Crouzy et al., 2016). Thresholds-based criteria were used to discriminate single pollen grains from debris, dust or agglomerates. In order to obtain a reasonable panel of the relevant pollen types, 60 calibrations were performed for 21 different taxa. Focus was set on repeating calibrations, if possible under varying conditions. Only fresh pollen was used and time between collection and calibration was reduced to a minimum (range: 15-120 minutes). The presence of agglomerates and debris was investigated 165 by collecting histograms of the optical size and of the fluorescence intensity of the recorded events. Cut-offs were introduced accordingly, in order to retain only single pollen grains. The device was also in Pollen mode, i.e. filtered out particles smaller than 5 µm of optical size.

### 2.3 Data analysis methods in Siauliai

Both modalities of the Rapid-E output (scattering image and the fluorescence spectra) were processed 170 independently with artificial neural networks (ANN) and the scores were merged to obtain the final classification result.

The scattering image has a peculiarity requiring special treatment. Depending on the particle position with regard to the laser beam while passing through it, the apparent size of the particle deduced from the Rapid-E scattering image varied for different recordings (Figure 1). Therefore, at the beginning, a characteristic 175 template of 44x20 pixels was identified. The template was used to localize the most similar areas in the scattering images. A convolutional neural network was trained for pollen recognition from such images. ANN for scattering images consists of two convolutional blocks for the feature extraction and two fully connected layers for classification (Figure 3). Every convolutional block consists of the 2D convolutional layer, the batch normalization layer, the ReLU activation layer, and the maxpooling layer. One mask of 180 the convolutional layer has size of 5x5. The convolutional layer of the first block has 16 filters, and the one of the second block has 32 filters. The maxpooling layer selects the maximal response from the area of 2x2. At the output of the second convolutional block, the size of the feature vector is 1760. The first fully connected layer has 256 neurons. The second fully connected layer classifies these vectors to the number of pollen classes chosen for the calibration. The ANN was trained using the cross-entropy loss. 185 The fluorescence analysis also starts from pre-processing. Similarly to the scattering image, particle position with regard to the laser beam influenced the features (first of all, intensity) of the fluorescence. Apart from that, the deep-UV laser is activated for a short period of time triggered by the 400 nm scattering image analysis, i.e. it can partly or fully miss the particle if fired at a wrong moment of time. As a result, only half of pollen grains recorded by the 400 nm laser produce reasonable intensity of the 190 fluorescence (Table 1).

The fluorescence spectrum was processed by a multilayer perceptron ANN (Figure 4).

Because the Rapid-E device in Siauliai often gives saturated short-wavelength fluorescence spectrum at the first time moment, only half of the possible wavelengths were included in the feature vector. Subsequent reduction of the light intensity allowed inclusion of all 32 values starting already from the 195 second time moment. Dropout and batch normalization layers are used for ANN regularization. This ANN





also was trained using the cross-entropy loss. Results of two ANN were fused by summing scores of every pollen type.

With both networks, care was taken to follow-up the training process and to avoid over-fitting the networks. Despite the large volume of the samples (Table 1), certain over-fitting was possible after many
training cycles – see the Discussion section.

## 2.4. Data analysis methods in Novi Sad

All Rapid-e signals (i.e. scatter, spectrum and life time) were transformed into images and jointly processed by a single ANN (Figure 5). Its architecture considers the same input dimensions of every image, and since the scatter signal could vary in the number of acquisitions, each image's width was
equalised by founding its centre of mass and either cutting or zero-padding to fit to 24x70 pixels. The dynamic range of each image was reduced by replacing each pixel value with its logarithm, which resulted in enhancing of the low intensity pixels. Images from temporally resolved spectrum data and all bands of the life time data were used unprocessed.

Only the data of sufficient strength were deemed suitable for the analysis. Experiments in Novi Sad
indicated that the threshold of the particle fluorescence intensity level to be > 1500 units at the Rapid-E scale for at least one emitted wavelength.

Since there are three sources of information, the data fusion was needed aiming to create an architecture that would allow the gradient to flow through the whole network, so that the back-propagation can be done updating the weights for each distinct source. Each input image in the ANN goes through its own
framework, consisting of 2D convolutional layers, replication padding layers, ReLU activation functions, batch normalization layers, max pooling and the dropout layers, combination of which we will call the convolutional block. The ANN was trained using NLL loss and the Stochastic Gradient Descent with the learning rate of 0.001 and the momentum of 0.9.

For scattering images, we used two convolutional blocks. The first convolutional layer had 10 filters with
the kernel size of 5x5, and the second one had 20 filters with the kernel size of 3x3. For the spectral images, the convolutional layer of the first block had 50 filters with the kernel size of 5x5, and the one of the second block had 100 filters with the kernel size of 3x3. For the lifetime images, the first convolutional layer had 70 filters with kernel size of 7x1, the second one had 140 filters with kernel size of 5x1 and the one of the final block had 200 filters with the kernel size of 3x3. At the output of the final convolutional
block, the sizes of the feature vectors for scattering image, fluorescence spectrum and lifetime are 1800, 1600 and 1400, respectively. Each feature vector is passed through one fully connected layer with 50 neurons. Those features were concatenated resulting in the feature vector of dimension of 150. The second and the final fully connected layer was of the size of the number of classes, after which the samples were classified with the log-softmax activation function.

## 230 2.5 Data analysis methods in Payerne: classifier and aspects related with operational use

The device outputs were first normalized using the respective maxima of the signals: the maximal scattering intensity, the maxima of each of the four lifetime bands and the maximal fluorescence. For scattering, the image was in addition centered and cut to a 24 x 100 shape. Only calibrations with optical





size above 10 micrometers were retained, and a fluorescence signal in a range and spectrum compatible
with single pollen grains was kept (see (Crouzy et al., 2016) for examples of spectra). The optical size
corresponding to 10 micrometers was estimated by comparing the integral of the scattering signal of 5
micrometer PSLs with the integral of the scattering signal for *Urtica* and *Parietaria* pollen grains. For
scattering, 5x5 convolutions were applied with 32 filters, ReLU activation, and the pooling layers
operated on a 2x2 window. For lifetime, 1D convolution was applied with ReLU activation, with windows
size 10x1 and with 10 filters. For the spectrometer, asymmetric 2x4 convolution was applied with 8 filters
with ReLU activation. The ANN was trained using the Adam optimizer and categorical cross-entropy as
loss function (Figure 6). In order to retain flexibility, additional features were inserted before the final
fully-connected layers: features computed from the raw signals indeed showed promising results in [30].
For the moment, only the maximum and the integral of the scattering together with the maxima of each
of the four lifetime bands and the maxima of the first three spectrum acquisition were used.

Even if high expectations on the performance of the classifier are met, problems are bound to occur in the
form of false-positive detections. Even a few percent of the error in discriminating between two pollen
types can lead to problematic drifts. For example, Birch pollen concentrations regularly exceed 1000
particles per cubic meter in Switzerland in spring. If just 2% of these are mis-interpreted as, e.g., Ambrosia
pollen, the false concentration of 20 grains per cubic meter would be already significant for allergy
analysis. In order to cope with this, we introduced two methods. Firstly, events with classifications below
a certain threshold were disregarded, as was done in (Crouzy et al., 2016) where a reduction of sampling
of 20% lead to an increase in precision of about 10%. Secondly, we require in operational monitoring at
least a few events with extremely good classification score to occur on the day considered or on the two
preceding days, this condition is applied uniformly over the pollen season to check which pollen taxa are
present in the air. We preferred this method over the expert supervision or calendar rules due to the fact
that it bases only on measurement.

## 3. Results obtained in Siauliai

The above technology was used to answer the following questions:
- can we identify different pollen genera using the Rapid-E data?
- can we identify different species within the same pollen genus?
- what is the recognition accuracy for the most-common pollen types in Lithuania?

The analysis was started from a semi-qualitative consideration of the fluorescence spectra, primarily
aiming at demonstration of the capabilities and the limitations of the approach and preliminarily assessing
the principal possibility to construct a reliable particle recognition algorithm.

### 3.1. Qualitative comparison of the fluorescence spectra of different pollen species

#### 3.1.1. Comparison of fluorescence spectra of different species of the same genus

The experiment included three genera, for which we collected pollen from different species (Table 1):
*Salix*, *Pinus*, and *Acer*. Their fluorescence spectra are shown in Figure 7, where the solid lines represent
the normalized mean spectrum and shadows show the standard deviation range. We also computed the


Standard Error of the Mean value and performed Student tests to evaluate the significance of the difference between the mean spectra. The uncertainties of the mean spectra were a fraction of a % leading to the statistically significant difference at all wavelengths for both *Pinus* and *Acer* mean spectra and even for some wavelengths of the *Salix* spectra.

Despite statistically significant differences between the mean spectra, the sample standard deviation (shadowed ranges in Figure 7) was quite large. Therefore, it was not possible to distinguish between *Salix alba* and *Salix fragilis*. The normalised spectra *of Pinus sylvestris* and *Pinus mugo* coincided at the maximum value of the amplitudes at the wavelength of 460 nm but the mean amplitude of the *Pinus sylvestris* spectrum was higher in short-wave range ($< 450$ nm). At the longer wavelenghts (480-550 nm)

the amplitude was higher for the *Pinus mugo* pollen. However, these differences were well inside the sample standard deviation. The difference between the species of the *Acer* genus was the most-pronounced and, even taking the sample variability into account, these were the ones that could be distinguished. The *Acer pseudoplatanus* spectrum showed higher amplitude than *Acer negundo* in the short-wave range and lower in the central part of the spectrum (400-520 nm).

Therefore, two out of three tested genera allowed, in principle, an inter-genus species classification using the pollen fluorescence spectrum. However, the differences between them were evidently too small for the multi-species algorithm considered in the current paper. Practical work was therefore left for the follow-up studies.

### 3.1.2 Comparison of fluorescence spectra of species of different genera

The study included 11 different pollen genera (Table 1), whose spectra are shown in Figure 8 for recordings at every 500 ns starting from the first pulse reaching the detector. For all species, the most intense fluorescence was observed for the wavelengths from 390 to 570 nm, with different locations of the maximum and with different amplitude. For example, the highest mean intensity of fluorescence was recorded for the *Artemisia* pollen: it exceeded >7000. Meanwhile, the amplitudes of *Betula* and *Quercus*

reached more than 4000. In all cases, the first pulse had a wider wavelength range than the subsequent ones. The amplitudes of already the second recording (500 ns from the first pulse) was close to zero for wavelengths longer than 600 nm.

In addition, Figure 8 shows that not only the intensity of the first signal between separate genera differs, but the shape of the second recording is also specific, which is significant for the identification of the

pollen morphotype. For example, the difference in fluorescence intensity of *Salix* pollen between the first and second signals was larger than for other tested taxa. Tests with *Festuca* pollen actually showed that, unlike all other species, the signal amplitude grows during the first 500 ns resulting in the absolute maximum intensity of the spectrum registered at the second recording, 500 ns after the fluorescence is induced.

The qualitative analysis of the data was continued by grouping the data according to similarity of the fluorescence spectrum of the first recording (Figure 9).

*Alnus*, *Corylus* and *Betula* plants are in one taxonomic family, and our results indicate that their pollen has a similar fluorescence spectrum. Interestingly, according to the similarities of the fluorescence spectra, *Quercus* pollen appeared in the same group with *Betulaceae*, although the maximum value of its

mean of the normalised spectrum was the lowest in the group.





Another group in which the pollen fluorescence curves have similar shapes also consists of pollen of woody plants: *Populus* and *Salix*. They also bloom at a similar time; therefore their precise identification is an important but, as seen from Figure 9, challenging task. The tested grass pollens form a separate group, which however also included pollen of the woody plant *Juniperus*. This group is characterised by the high mean amplitude in short (< 400 nm) wavelength range.

## 3.2 Recognition skills

The key practical question for the Rapid-E application in the daily pollen monitoring is the accuracy of the pollen type classification, which we presented below via the confusion matrices. In these matrices, rows represent the actual type of pollen and columns are the assigned type. All values are in %, the sum of values over each row is 100%: every pollen has to be assigned to some type.

Rapid-E provided the scattering and fluorescence data arrays. The recognition procedure in Siauliai was built independently for each of them with subsequent fusion of the results (Tables 2-4).

### 3.2.1. Recognition using scattering images only

In the confusion Table 2, the classification is based on the scattering images. The overall recognition accuracy was at an unimpressive level of 44%.

Only very large and specific *Pinus* pollen was recognised correctly in 76% cases. In 13% of cases it was mis-classified as *Festuca*. The accuracy similar to the second-best species *Artemisia* was obtained for recognising the *Festuca* morphotype. *Corylus*, *Alnus* and *Betula* pollen were frequently confused inside the *Betulalceae* family. *Salix* pollen was confused with the *Populus* pollen, which identification accuracy is the lowest in this test. Other pollen morphotypes tested in the experiment are identified correctly in less than 50% of cases.

### 3.2.2. Recognition using fluorescence spectra

The results of pollen identification using the fluorescence spectrum are better than those of the scattering-based recognition. The total accuracy reaches 67% and, as seen from Table 3, pollen of *Pinus*, *Artemisia*, *Acer*, *Festuca, Juniperus* and *Salix* are well distinguished.

Grouping the genera, one can notice that the highest percentage of confusion was again within the *Betulaceae* family. *Betula* was confused with *Alnus* in 29% of cases. Due to confusion with *Alnus* and *Corylus*, about a fifth of the *Quercus* pollen was not recognized correctly. *Populus* could also be attributed to a multitude of pollen types. In particular, in 17% cases it was attributed to *Salix* and in 9% - to *Artemisia*.

The individual species of *Betulaceae* family were the worst identifiable from their fluorescence but compared to recognition from the scattering image (Table 2) the results are better. An analogous situation was in the case of *Salix*: identification from the fluorescence spectrum was more accurate than that based on the scattering images. The method also significantly improved identification of *Juniperus* and *Acer* pollen in comparison with the scattering images: 36% and 84%, 29% and 84% respectively.



### 3.2.3. Combined identification

Table 4 presents the outcome of the combined identification using both the scattering image and the fluorescence spectra. With the exception of *Alnus*, the combination of the identification methods showed better recognition skills than each of the methods separately.

Overall, 6 out of 11 tested pollen genera were identified with the accuracy better than 75%. The best results (> 91% of correct classification) were achieved for *Pinus* pollen. The pollen of the plants of *Betulaceae* genus was also less confused than with the separate methods, but the recognition of the individual species of this family was still poor. The fusion of scattering image and fluorescence algorithms significantly reduced confusion of *Festuca* and *Pinus* pollens. A particular improvement was obtained for

*Acer* and *Juniperus* (> 50%) comparing to the scattering-based classification. Gain over the fluorescence-only method was the largest for *Quercus*, which recognition improved by 14%. The identification of other pollen types improved by 2-10%.

Overall, the improvement over individual methods was ~23% compared to scattering images and ~7% compared to fluorescence.

# 4 Discussion

### 4.1. Comparison of the confusion tables obtained in Novi Sad, MeteoSwiss (Payerne) and Siauliai.

The study in Novi Sad was performed for the largely similar set of species using practically identical experimental part but independent analytical procedures and different pollen material for calibration. Although it is difficult to make exact comparison the confusion tables between the studies, it still sheds

some light on the overall performance and also clearly highlights the similarities and differences between the regions. Comparing the Tables 4 and 5, one can see that the difference in the recognition quality is about 10% for most of species, being practically identical for *Betula* (~50% in both studies) and *Quercus* (~60%). Somewhat higher skills in Novi Sad were obtained for *Corylus*, *Alnus* and *Populus* while in Siauliai higher skills were reached for *Acer* and *Artemisia*. It is interesting to note that the confusion

between the two chosen grass pollen morphotypes in Novi Sad was not notable and for these genera the Rapid-E data have certain discrimination potential. One can therefore conclude that the multi-species discrimination algorithms applied in these studies performed in a very similar way. It should be stressed however that the training of the ANNs were completely independent and used the local pollen grains. Therefore, the similar recognition quality does not imply similar pollen in these regions.

As mentioned earlier (Section 2.2.3), the calibration procedure used at MeteoSwiss was slightly different than in Novi Sad and in Siaulai. In addition, the focus was set at MeteoSwiss towards testing the ability to monitor relevant taxa. As a consequence, only a subset of the 60 calibrations performed was used to train the classifier. Taxa with a high relevance for monitoring or for which very good calibrations were available were selected. It was indeed observed that increasing the number of taxa could worsen the

problem of false positive detections (see below). An optimum for monitoring purposes was found when using 10 taxa. The performance of the corresponding classifier is shown in Table 6. It is interesting to note that, as expected, most errors occur within the *Betulacae* family, with an extremely low recall for *Alnus*. Note that this confusion table should be understood as a measure of the ability of the classifier to



distinguish between calibrations. As such, although calibrations were repeated, the classifier may to some
extend recognize the conditions under which the calibration was performed and the quality of the sample. Obtaining a classifier working only on the generic features of the taxa is a very difficult task. A holistic validation procedure, going from the analysis of device raw outputs (Section 3.1) analysis to the comparison with reference measurements (Section 4.5), is therefore essential.

## 4.2. Over-training – a problem?

The problem of potential over-training was addressed from two directions: via the standard training – vs – test datasets evaluation, and via an explicit verification of homogeneity of the datasets.

### 4.2.1. Performance in the training and test datasets

Prior to starting the ANN training, all datasets were split to the training and test subsets. The test subset in Siauliai consisted of 1000 particles picked at the end of every calibration event while all other particles
were used for training. The Siauliai ANN training continued until saturation of the recognition quality for the training dataset (see example in Figure 10), thus including the overfitting range. The maximum performance of the fluorescence-based recognition was obtained at the epoch of ~900, after which the over-fitting gradually picked up. Therefore, the ANN parameters after this epoch were taken as the study outcome. For the scattering-image-based training, a similar consideration suggested the epoch 3500 as
the optimum.
For Novi Sad (Figure 11), the training was stopped before the overfitting picked up and thus the parameters of the last trained epoch 3000 were used.

### 4.2.2. Test of homogeneity of the calibration datasets

One of the concerns regarding the fluorescence-based technology is the stability of the spectra for
different conditions of pollen grains, which are affected by ambient humidity, temperature, time they spent in the air, etc. Full-scale evaluation of this problem lies beyond the scope of this paper. Here, we only present a brief check demonstrating that this is not the major issue.
As stated in the methodological section, the calibration set for each pollen type consisted of up to 8 independent calibration sessions, sometimes performed in different days and with pollen of different age.
A simple check of homogeneity of the fluorescence spectra is then to use the data of one of these days as the training set and those from another day as the test subset. Substantial difference in the recognition quality would point at the inhomogeneous data.
This experiment was performed for only 4 species, which had such multi-days calibration sets. Therefore, the problem was significantly simpler: to distinguish only between these 4 species instead of 11 in the
above Siauliai tables 2-4. However, the important part was the difference between the training and test recognition quality.
Comparing the upper and lower rows of Figure 12, one can see that for the above epochs (3500 for scattering- and 900 for fluorescence-based ANNs), the quality of recognition for the training subset (one day) and test subset (another day) differ by <5% for all 4 species. Therefore, we conclude that the
conditions during the different days of calibration did not affect the homogeneity of the dataset.





### 4.3. Comparison with other studies on pollen recognition

During recent years, a number of attempts to obtain information about pollen concentration in the air in real time have been undertaken. However, even the most-successful tests carried out with Hund BAA500 (Oteros et a., 2015), Yamatronics KH-3000-01 [34] (Kawashima et al., 2017), and Plair PA-300 (Crouzy et al., 2016) devices, strongly advancing the pollen monitoring field, left open the questions of scalability and replicability of the results. They also did not touch the topics related to application of the tested systems in the operational context. Application of yet-another new device – Plair Rapid-E – in our study was pursuing, apart from the scientific objectives, the operational implementation as a mid- to long-term goal. However, having tested 14 different pollen morphotypes, we found that significant work is still needed.

One of the challenges to the automatic monitors is the rich mixture of pollen types in Europe that all pose significant allergenic threat. This makes it particularly difficult for the monitors to satisfy the needs of allergic people and allergologists – unlike many other regions. Thus, *Cryptomeria japonica* is the species that has been identified in the automated pollen identification system more than 10 years ago by Kawashima et al. (2007) and is still the main pollen type recognised by that system (Kawashima et al., 2007; Wang et al., 2013; Wang et al., 2014; Takahashi et al., 2018). However, it seems to be more or less sufficient for that region.

Varying level of allergenicity of species within a single genus or a family raises the question if the intra-genus classification is possible. Hirst-based manual techniques do not allow it: pollen grains are too similar in the microscopic analysis. Our results show that such level of identification is not immediately possible using Rapid-E information either – at least the multi-species discrimination algorithm is not sufficiently sensitive. In particular, our data demonstrated that the fluorescence spectra of the *Salix alba* and *Salix fragilis* species were all but identical. More promising were the experiments with *Pinus* and *Acer* (Figure 4) and some grasses (Table 5) where the work should be continued with different identification algorithms built for these very species after their separation from other pollen types. Other genera should also be tested.

We also found out that the fluorescence spectra can be similar not only between species of a particular genus but also between different families. We found several groups of otherwise unconnected species, whose spectra are similar to a degree that does not allow their reliable differentiation with the first-level multi-species algorithm (Figure 9). Similar results were obtained in the studies conducted by D. J. O'Connor with co-authors (2011). They assessed the fluorescence spectrum of pollen of *Betulaceae* family and stated that "birch and alder spectra closely resemble each other although there is a possibility that the birch pollen is less fluorescent than alder". Our results show that in the case of *Alnus*, the fluorescence amplitudes are higher than of *Betula*, but the spectra are indeed similar.

Similar spectra of *Salix* and *Populus* pollen (Figure 7) also resulted in poor differentiation between them. At the same time, the degree of confusion was higher for *Populus* than for *Salix*, which was recognized much better than *Populus* (Tables 2 – 4). This is in agreement with results obtained with Hund BAA500 by Oteros et al. (2015), who identified *Salix* pollen as the worst of all pollen types analysed in their work (Oteros et al., 2015). The BAA500 algorithm is based on recognition of the particle shape, which can be weakly related to the scattering images in our study – the very part that showed substantial confusion of almost all studied pollens with *Salix* and *Populus*.



Crouzy et al. suggested that a non-zero fluorescence amplitude around 600 nm wavelength is incompatible with pollen from the *Betulaceae* family (*Alnus*, *Carpinus*, *Corylus* and *Betula*) but could possibly be observed for grass pollen (*Dactylis* and *Phleum*) (Crouzy et al., 2016). Our results support this suggestion
and in addition the test in Novi Sad shows that ANN could show some discriminatory power between *Dactylis* and *Cynodon*. Noteworthy, recognition of the herbaceous plants (*Festuca*, *Artemisia*) was considerably better than that of pollen of *Betulaceae* family also in Siauliai (Table 4).

In general, our results strongly suggest that combination of recognitions based on scattering images and fluorescence spectra have the highest potential as they exploit very different features of the pollen grains
and can serve as complementary methodologies. This approach showed the highest overall recognition accuracy exceeding 70%. The lifetime of the fluorescence was explicitly included as a separate set of variables in Novi Sad and Payerne and implicitly used in Siauliai via incorporation of the spectra taken in different moments.

One can note that the above recognition accuracy is in an apparent contradiction with published results
(Crouzy et al., 2016), where the skills were significantly higher: 91% obtained with PA-300. However, there are several important differences between the approaches. Firstly, the pre-filtering of the particles is substantially stricter in the procedure of Crouzy et al. (2016) than in Siauliai and Novi Sad and about 20% of outputs were filtered out as uncertain (threshold of the classification quality). Secondly, the accuracy of the recognition depends significantly on the number of pollen morphotypes used for the test
(8 by Crouzy et al). In an extreme case, automated discrimination of just one species (*Cryptomeria japonica*) from non-pollen particles using KH-3000 was high already 10 year ago (Kawashima et al., 2007; Kawashima et al., 2017). Similarly, the high fraction of BAA500 true positive counts (93,3%) against manual analysis of individual species by Oteros et al. (2015) went down to 65% as in our study when the recognition of 13 pollen morphotypes was requested. It took an additional training of the
algorithm to raise it up to the same 72%. Finally, it should also be noted that PA-300 delivers fewer parameters than Rapid-E, possibly making it difficult to identify the important combinations in the raw signal in a single-level many-to-many identification task. Application of additional levels of the discrimination filters can substantially improve the results.

## 4.4. Possible ways to improve the recognition skills

The dependence of the recognition quality on the number of categories is one of the directions of the future research. It may be possible to consider independent groups of pollens that never (or very rarely) appear in the air at the same time – but it can make the algorithm place-specific. It is vital however to obtain improvement of the algorithm for reliable separation of pollens that can be in the air together (e.g., *Betulaceae*, *Quercus*, and the like).

Considering improvements of the recognition algorithms, Matsuda and Kawashima (2018) suggested the "extract window" method of analysis of the scattering images, which enabled to distinguish unique ranges of light scattering intensities for each taxon of pollen of that study but the reliability of the algorithm is known only for 5 pollen morphotypes. Development of this and similar approaches for the Rapid-E scattering images may eventually improve this line of analysis and, subsequently, push up the overall
scores.





Since the output of the ANN can be transformed to give a vector of probabilities, where each element $i$ of the vector represents the probability that the sample belongs to class $c_i$, we expect improvement of the classification accuracy if we demand that the classification occurs only if the highest probability in that vector is greater than some probability threshold, but with the price of discarding the samples below the defined threshold. This direction was initially explored by Crouzy et al (2016) and showed high potential: discarding 20% of samples led to an increase of precision of about 10%. The future studies will encompass this challenge of losing samples while introducing probability threshold.

As a more radical approach, one can challenge the solo usage of ANNs without a-priori relations derived from physical or chemical features of each pollen type. Even generic considerations of scattering and fluorescence theories might hint on quantities, which show enhanced contrast in comparison with the raw data.

## 4.5 Lessons from the comparison of the Hirst and Rapid-E measurements

Comparison of the Rapid-E of MeteoSwiss with the operational Hirst measurements in Payerne from February to June 2018 extended the results of Crouzy et al. (2016) to more important taxa (Figure 13), but also showed that robust determination of the sampling still needs to be achieved. In order to obtain pollen concentrations, large particles presenting bimodal fluorescence spectra with position and intensity of maxima compatible with the observations made from calibrations (see Section 3.1) were first selected. Then, the classifier presented in Section 2.5 was applied. The effective sampling of Rapid-E is the result of a series of physical and algorithmic processes: the sampling efficiency of the Sigma-2 head, the imperfect targeting by lasers and the drop-offs due to below-the-threshold signal during the classification step. In Figure 13, the Rapid-E data are scaled with species-dependent factors (constant over the season) bringing the seasonal mean to that of the Hirst time series. The issue deserves attention since, as shown by the Novi Sad results, tightening the thresholds improves the recognition skills but increases the drop-offs at the recognition stage. Sampling with *Poaceae* is the highest, *Pinaceae* present a 2% decrease of sampling and *Betula* presents a 33% decrease in sampling with respect to *Poaceae*. False positive are a significant issue with *Fraxinus*: due to the necessary activation thresholds sampling is dramatically reduced (75%). As a consequence of those limitations, the results presented here should not be taken as a complete demonstration of operational capabilities.

The suppression of the false-positive detections as described in Section 2.5 worked quite efficiently but still an evident false-positive event resulting from the *Betula* misinterpretation as *Poaceae* is visible in the beginning of April. Further work is required to completely remove such events, and, as a last resort, expert supervision could be used in an operational setup.

## 4.6. Opinion of the Rapid-E producer

During the work, we have been in periodic contact with the Plair company regarding features and issues of the Rapid-E devices used by our groups. Having the paper finalised, we asked their feedback.

D.Kiselev, Plair: "Our impression concerning the presented material is mixed. While I see some positive and encouraging results, my main critics would be addressed to you*r* calibration sets, which cleaning and filtering falls short of the actual needs. Our results are 5-10 % better without overfitting the data or other



special processing. Time series Plair gets for "problematic pollens" like Betula, Corylus and Alnus are
actually very good. Our goal is to obtain high quality time series calculated in real-time by the instruments
and the good calibration is essential for that."
We agree with importance of the calibration datasets; the procedures ensuring their quality are described
in the Methodology section and further explained in the discussion above. Noteworthy, our groups were
working largely independently using local pollen and original methods of the data collection and
processing. Therefore, the similarities in the observed features provide additional support for our
conclusions. Unfortunately, the Plair company declined to reveal any detail of procedures and datasets
substantiating their message. Therefore, independent evaluation of that algorithm against the common
criteria described in this paper was not possible.

## 5 Conclusions

We conducted the first analysis of the pollen monitoring capabilities of the new automatic pollen detector
Rapid-E. Using the very limited data pre-processing and basic ANN classification it was shown that, if
comparatively large number of pollen types is considered, stand-alone scattering- and fluorescence- based
recognition algorithms fail to produce reliable results for majority of species. The combination of these
algorithms performed better exceeding 80% accuracy for 5 out of 11 species. Therefore, this combination
can be considered as the first-stage classification of pollen types. It should be followed by more in-depth
discrimination efforts, including also life time of florescence into the classification model, etc.
The fluorescence spectra showed similarities among several tested species ending up with three groups:
(*Alnus*, *Corylus*, *Betula* and *Quercus*), (*Salix* and *Populus*), and (*Festuca*, *Artemisia*, *Juniperus*) – as
identified from the Siauliai data. The classification between the groups was comparatively easy and
reliable but distinguishing pollens inside the groups turned out more problematic.
Attempts to distinguish between the species of the same genus showed certain potential for some genera
but more work is needed.
The results obtained in Siauliai and Novi Sad with very similar experimental setup but independent
analysis, showed comparable results confirming the overall conclusions. They also pointed out at certain
limitations of replicability of the raw data features between the devices, which will require an additional
conversion step to make them compatible. In this line, the comparison performed at MeteoSwiss shows a
reasonable potential for automatic monitoring of important taxa, however it is not clear to which extent
algorithms can be transposed from one device to another.
The in-depth discussion and improvement of the methodology and the extension to more taxa goes beyond
the scope of this paper. We decided to communicate early the current results, as well as the methods
developed independently by three teams currently working with Rapid-E counters, in order to stimulate
parallel developments by the user community of the Rapid-E devices. The emergence of such community
is a good opportunity to address generalization and replicability of the device-specific results. We also
believe that moving from expert supervision or calendar methods to the approach presented here basing
only on device outputs for, e.g., elimination of false-positive detections could be of help for other
automatic monitoring systems.
Among the main challenges to be resolved in the future work, the most important ones are:
-  to obtain reliable recognition skills at least for the pollen types that can be in the air at same time




- to reach full replicability of the algorithms and results across the different copies of the same monitors
- to resolve specific questions related to the algorithm construction and training – including the minimal sample volume, problems of over- and under-fitting, preprocessing and pre-filtering of the data, false-positive identifications, etc.

Successful resolution of these questions will open the way for wide applications of the automatic particle counters for pollen observations.

**Code and data availability**

All data and algorithms presented in the paper are experimental and subject to further development. They are available for research purposes on-request basis from the authors of the manuscript. Work is in progress to harmonise the algorithms and make them public together with the data via open software and data repositories. Possibility of GPL-type license is being evaluated.

**Author Contributions:**

All the authors made significant contributions to this study. Conceptualization M.S.; Writing – Original Draft Preparation, Review & Editing I.Š., M.S., L.Š, B.S., B.C., B. Cr.; Methodology G.V., I.Š, L.Š; Data analysis and visualization G.D; L.V; M.S; Experiment in Novi Sad P.M., S.B., M.P., B.S; Experiment in Payerne B.C., B. Cr.

**Funding**: This research is funded by the European Social Fund according to the activity "Improvement of researchers' qualification by implementing world-class R&D projects" of Measure No. 09.3.3-LMT-K-712-01-0066. Support of the PS4A project of Academy of Finland is kindly acknowledged. The experiment in Novi Sad was performed in the frame of RealForAll project (2017HR-RS151) co-financed by the Interreg IPA Cross-border Cooperation programme Croatia – Serbia 2014-2020 and Provincial secretariat for Science, Autonomous Province Vojvodina, Republic of Serbia (contract no. 102-401-337/2017-02-4-35-8).

**Acknowledgements**

Authors thank the Plair engineering team for their advices concerning the Rapid-E operations.

**Conflicts of Interest:** The authors declare no conflict of interest.

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



**Table 1.** Pollen used for testing the identification capabilities of the instrument in Siauliai

| Plant group | Total particles counted by 400 nm laser | Fluorescent particles * |
|---|---|---|
| *Festuca* | 21808 | 12205 |
| *Artemisia* | 15521 | 13370 |
| *Corylus* | 14858 | 10865 |
| *Alnus* | 13692 | 10486 |
| *Betula* | 20676 | 12089 |
| *Salix alba* | 15383 | 13431 |
| *Salix fragilis* | 12942 | 10401 |
| *Populus* | 15340 | 10963 |
| *Acer negundo* | 11832 | 8647 |
| *Acer pseudoplatanus* | 11030 | 7372 |
| *Juniperus* | 17926 | 10404 |
| *Quercus* | 17677 | 8934 |
| *Pinus sylvestris* | 14224 | 8537 |
| *Pinus mugo* | 13399 | 8287 |

\* the particle fluorescence intensity level> 1500 at the Rapid-E scale for at least one emitted wavelength


**Table 2.** Confusion table for pollen taxa identification by using ANN based on scattering image.
Multiclass accuracy 44 %

| Plant genus | *Festuca* | *Artemisia* | *Corylus* | *Alnus* | *Betula* | *Salix* | *Populus* | *Acer* | *Juniperus* | *Quercus* | *Pinus* | |
|---|---|---|---|---|---|---|---|---|---|---|---|---|
| *Festuca* | **52** | 1 | 1 | 2 | 1 | 3 | 3 | 5 | 5 | 9 | 19 | |
| *Artemisia* | 2 | **58** | 1 | 1 | 4 | 13 | 3 | 4 | 9 | 3 | 2 | |
| *Corylus* | 3 | 1 | **38** | 20 | 20 | 1 | 1 | 5 | 1 | 9 | 1 | |
| *Alnus* | 4 | 3 | 21 | **29** | 19 | 3 | 1 | 6 | 1 | 11 | 2 | |
| *Betula* | 4 | 4 | 19 | 15 | **37** | 3 | 1 | 5 | 1 | 9 | 2 | |
| *Salix* | 5 | 8 | 1 | 0 | 5 | **51** | 13 | 3 | 6 | 7 | 1 | True label |
| *Populus* | 16 | 4 | 0 | 1 | 1 | 16 | **28** | 4 | 16 | 8 | 6 | |
| *Acer* | 8 | 5 | 6 | 7 | 4 | 4 | 5 | **29** | 4 | 23 | 5 | |
| *Juniperus* | 16 | 19 | 0 | 0 | 0 | 4 | 10 | 2 | **36** | 3 | 10 | |
| *Quercus* | 7 | 2 | 10 | 6 | 3 | 3 | 3 | 14 | 2 | **48** | 2 | |
| *Pinus* | 13 | 2 | 0 | 1 | 1 | 1 | 1 | 1 | 3 | 1 | **76** | |

Predicted label






**Table 3.** Confusion table for pollen taxa identification using fluorescence spectrum.
Multiclass accuracy 67 %

| Plant genus | Festuca | Artemisia | Corylus | Alnus | Betula | Salix | Populus | Acer | Juniperus | Quercus | Pinus |  |
|---|---|---|---|---|---|---|---|---|---|---|---|---|
| Festuca | **84** | 2 | 0 | 0 | 0 | 0 | 1 | 5 | 3 | 0 | 5 | |
| Artemisia | 2 | **80** | 0 | 0 | 0 | 1 | 3 | 6 | 8 | 0 | 0 | |
| Corylus | 2 | 0 | **53** | 15 | 7 | 6 | 1 | 0 | 1 | 14 | 1 | |
| Alnus | 1 | 0 | 14 | **45** | 19 | 3 | 4 | 1 | 1 | 10 | 2 | |
| Betula | 3 | 1 | 7 | 29 | **38** | 3 | 4 | 0 | 1 | 12 | 2 | |
| Salix | 0 | 2 | 7 | 2 | 3 | **72** | 9 | 0 | 1 | 4 | 0 | True label |
| Populus | 2 | 9 | 3 | 2 | 2 | 17 | **53** | 4 | 4 | 3 | 1 | |
| Acer | 6 | 3 | 1 | 0 | 1 | 0 | 2 | **84** | 2 | 1 | 0 | |
| Juniperus | 6 | 5 | 0 | 0 | 1 | 0 | 3 | 0 | **84** | 0 | 1 | |
| Quercus | 1 | 0 | 12 | 13 | 9 | 6 | 2 | 0 | 1 | **55** | 1 | |
| Pinus | 11 | 0 | 0 | 0 | 1 | 0 | 0 | 0 | 1 | 1 | **86** | |

Predicted label

**Table 4.** Confusion table based on two ANNs fused by summing scores of every pollen type
Accuracy 73 %

| Plant genus | Festuca | Artemisia | Corylus | Alnus | Betula | Salix | Populus | Acer | Juniperus | Quercus | Pinus |  |
|---|---|---|---|---|---|---|---|---|---|---|---|---|
| Festuca | **88** | 1 | 0 | 0 | 0 | 0 | 2 | 5 | 2 | 0 | 2 | |
| Artemisia | 2 | **86** | 0 | 0 | 0 | 2 | 1 | 4 | 5 | 0 | 0 | |
| Corylus | 2 | 0 | **63** | 17 | 8 | 1 | 0 | 0 | 0 | 9 | 0 | |
| Alnus | 1 | 0 | 15 | **53** | 18 | 2 | 1 | 0 | 0 | 9 | 1 | |
| Betula | 3 | 1 | 9 | 30 | **47** | 1 | 1 | 0 | 1 | 6 | 1 | |
| Salix | 1 | 1 | 2 | 1 | 2 | **78** | 10 | 0 | 1 | 4 | 0 | True label |
| Populus | 3 | 6 | 1 | 1 | 1 | 18 | **58** | 3 | 3 | 5 | 1 | |
| Acer | 5 | 2 | 1 | 1 | 1 | 0 | 2 | **86** | 1 | 1 | 0 | |
| Juniperus | 4 | 4 | 0 | 0 | 0 | 0 | 3 | 1 | **87** | 0 | 1 | |
| Quercus | 2 | 0 | 9 | 10 | 5 | 4 | 1 | 0 | 0 | **69** | 0 | |
| Pinus | 7 | 0 | 0 | 0 | 0 | 0 | 0 | 0 | 1 | 0 | **91** | |

Predicted label






**Table. 5.** Confusion table obtained in Novi Sad. Accuracy 74% (obs different number of species)

| Plant genus | *Dactilis* | *Cynodon* | *Corylus* | *Alnus* | *Betula* | *Salix* | *Fraxinus* | *Populus* | *Acer* | *Artemisia* | *Taxus* | *Quercus* | *Picea* | *Cedrus* | |
|---|---|---|---|---|---|---|---|---|---|---|---|---|---|---|---|
| *Dactilis* | **78** | 3 | 0 | 3 | 0 | 2 | 0 | 4 | 5 | 0 | 1 | 0 | 0 | 0 | |
| *Cynodon* | 4 | **70** | 0 | 0 | 0 | 6 | 1 | 0 | 0 | 12 | 5 | 0 | 2 | 0 | |
| *Corylus* | 0 | 0 | **64** | 6 | 12 | 1 | 10 | 0 | 0 | 0 | 4 | 2 | 0 | 0 | |
| *Alnus* | 1 | 2 | 6 | **72** | 3 | 2 | 3 | 2 | 1 | 2 | 6 | 3 | 0 | 0 | |
| *Betula* | 1 | 0 | 25 | 5 | **51** | 3 | 3 | 0 | 0 | 1 | 1 | 10 | 1 | 0 | |
| *Salix* | 3 | 1 | 0 | 2 | 1 | **80** | 3 | 2 | 2 | 2 | 1 | 3 | 1 | 0 | True label |
| *Fraxinus* | 0 | 0 | 7 | 1 | 3 | 2 | **79** | 1 | 0 | 0 | 4 | 3 | 0 | 0 | |
| *Populus* | 5 | 3 | 1 | 4 | 0 | 4 | 3 | **71** | 1 | 1 | 3 | 4 | 1 | 0 | |
| *Acer* | 8 | 1 | 0 | 2 | 1 | 4 | 0 | 0 | **73** | 0 | 1 | 9 | 1 | 0 | |
| *Artemisia* | 1 | 5 | 0 | 2 | 1 | 4 | 0 | 1 | 0 | **84** | 0 | 1 | 1 | 0 | |
| *Taxus* | 0 | 3 | 0 | 2 | 0 | 0 | 0 | 0 | 0 | 0 | **93** | 1 | 1 | 0 | |
| *Quercus* | 1 | 0 | 4 | 5 | 8 | 9 | 2 | 1 | 1 | 1 | 1 | **63** | 4 | 0 | |
| *Picea* | 3 | 3 | 0 | 2 | 4 | 3 | 0 | 1 | 1 | 7 | 1 | 13 | **61** | 0 | |
| *Cedrus* | 0 | 0 | 0 | 0 | 0 | 1 | 0 | 1 | 0 | 1 | 0 | 0 | 1 | **95** | |

Predicted label

**Table 6** Confusion table obtained at MeteoSwiss, Payerne. Accuracy 80% (obs different number of species)

| Plant genus | *Alnus* | *Betula* | *Carpinus* | *Cupressus* | *Fagus* | *Fraxinus* | *Corylus* | *Pinus* | *Poaceae* | *Taxus* | |
|---|---|---|---|---|---|---|---|---|---|---|---|
| *Alnus* | **27** | 27 | 1 | 0 | 0 | 1 | 43 | 0 | 0 | 1 | |
| *Betula* | 1 | **83** | 2 | 0 | 0 | 4 | 7 | 0 | 0 | 1 | |
| *Carpinus* | 0 | 13 | **74** | 0 | 0 | 2 | 3 | 0 | 6 | 1 | |
| *Cupressus* | 0 | 3 | 1 | **84** | 0 | 0 | 1 | 2 | 1 | 8 | |
| *Fagus* | 0 | 2 | 3 | 1 | **88** | 0 | 1 | 1 | 2 | 3 | True label |
| *Fraxinus* | 0 | 12 | 2 | 0 | 0 | **78** | 2 | 1 | 2 | 3 | |
| *Corylus* | 4 | 8 | 0 | 0 | 0 | 0 | **87** | 0 | 0 | 0 | |
| *Pinus* | 0 | 0 | 0 | 0 | 0 | 0 | 0 | **98** | 0 | 2 | |
| *Poaceae* | 0 | 3 | 8 | 1 | 0 | 1 | 0 | 1 | **82** | 4 | |
| *Taxus* | 0 | 0 | 0 | 1 | 0 | 0 | 0 | 1 | 0 | **97** | |

Predicted label






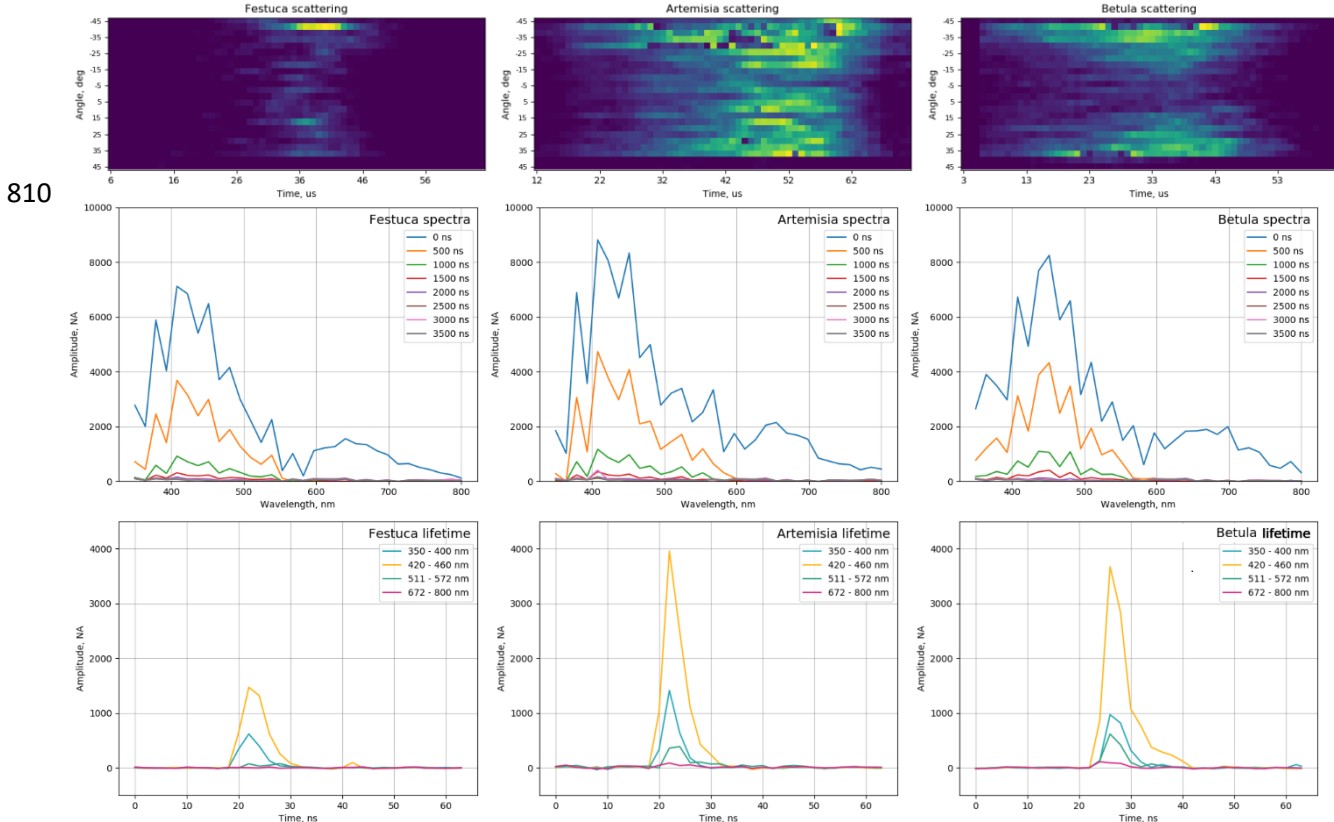

**Figure 1.** Examples of scattering images, fluorescence spectra and lifetimes of selected pollen types

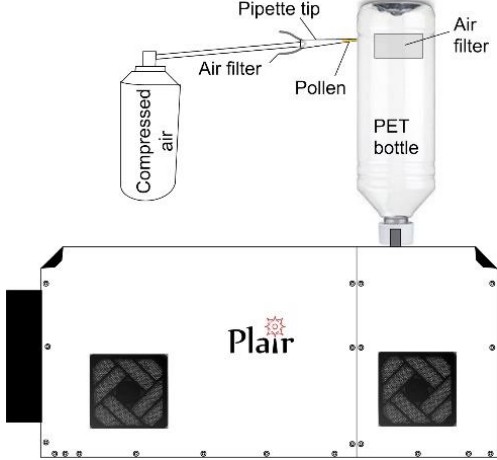


**Figure 2.** The scheme of the experiment for identification of pollen





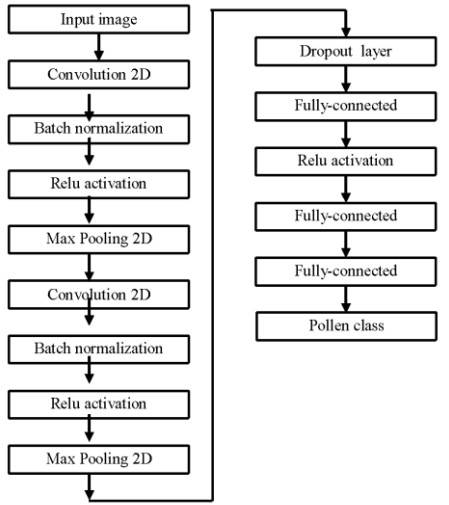

**Figure 3.** Neural network for pollen classification from the scattering image

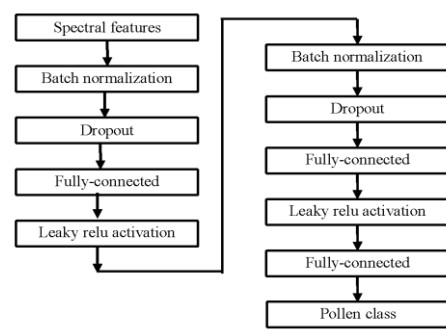

**Figure 4.** Neural network for pollen classification by fluorescence spectrum

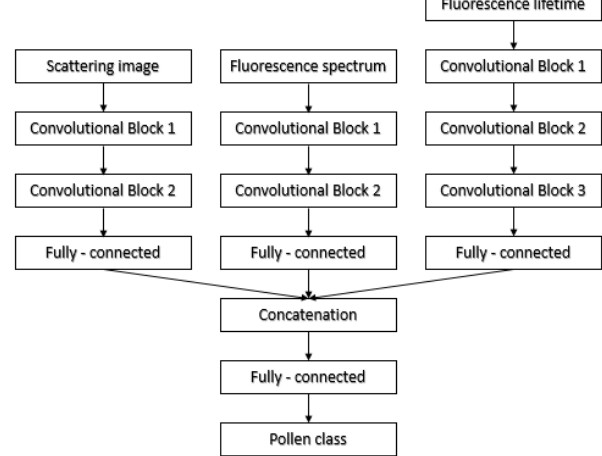

**Figure 5.** Neural network for pollen classification by all three signals




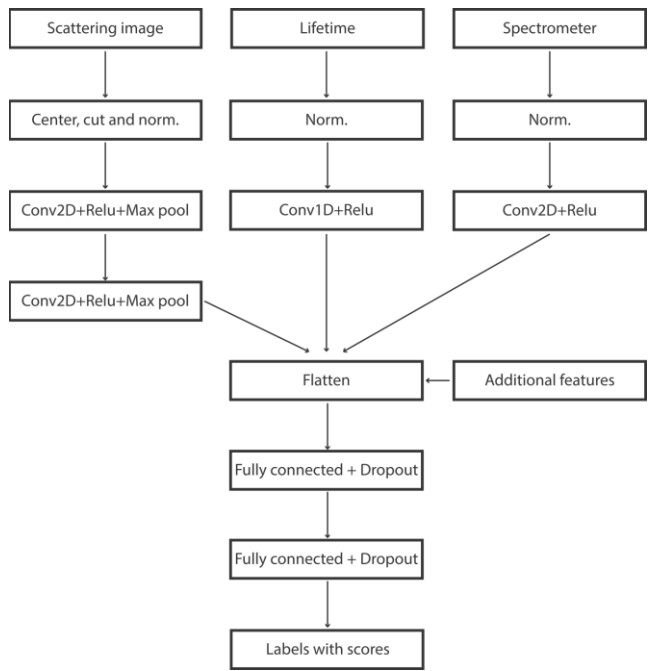

825            **Figure 6.** Neural network used for classification at MeteoSwiss





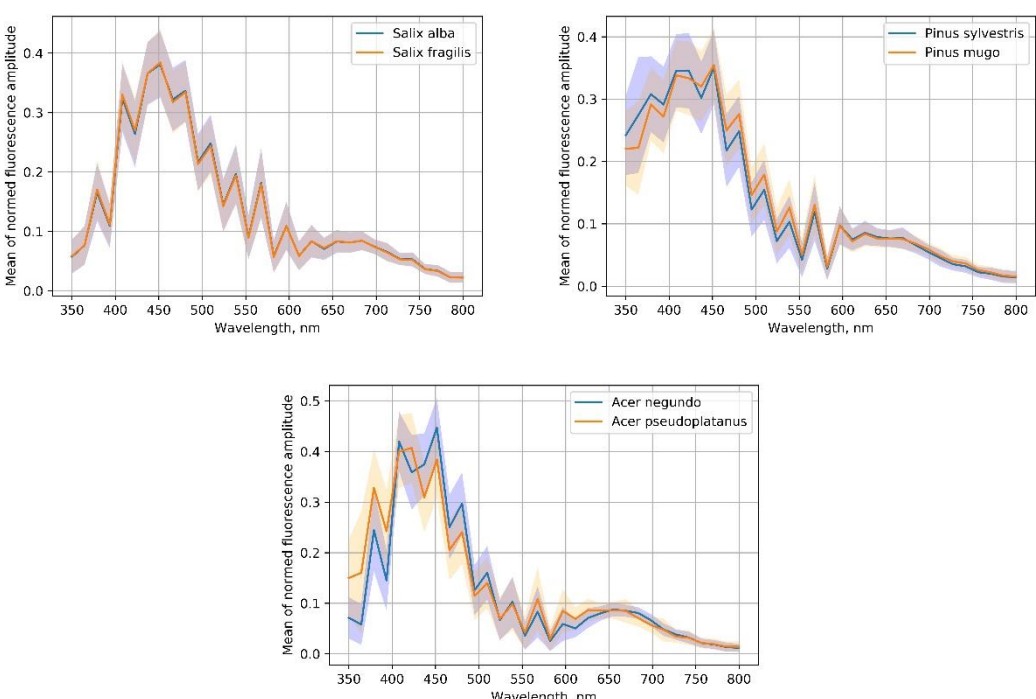

**Figure 7.** Fluorescence spectra (first acquisition) of *Salix*, *Pinus* and *Acer* species




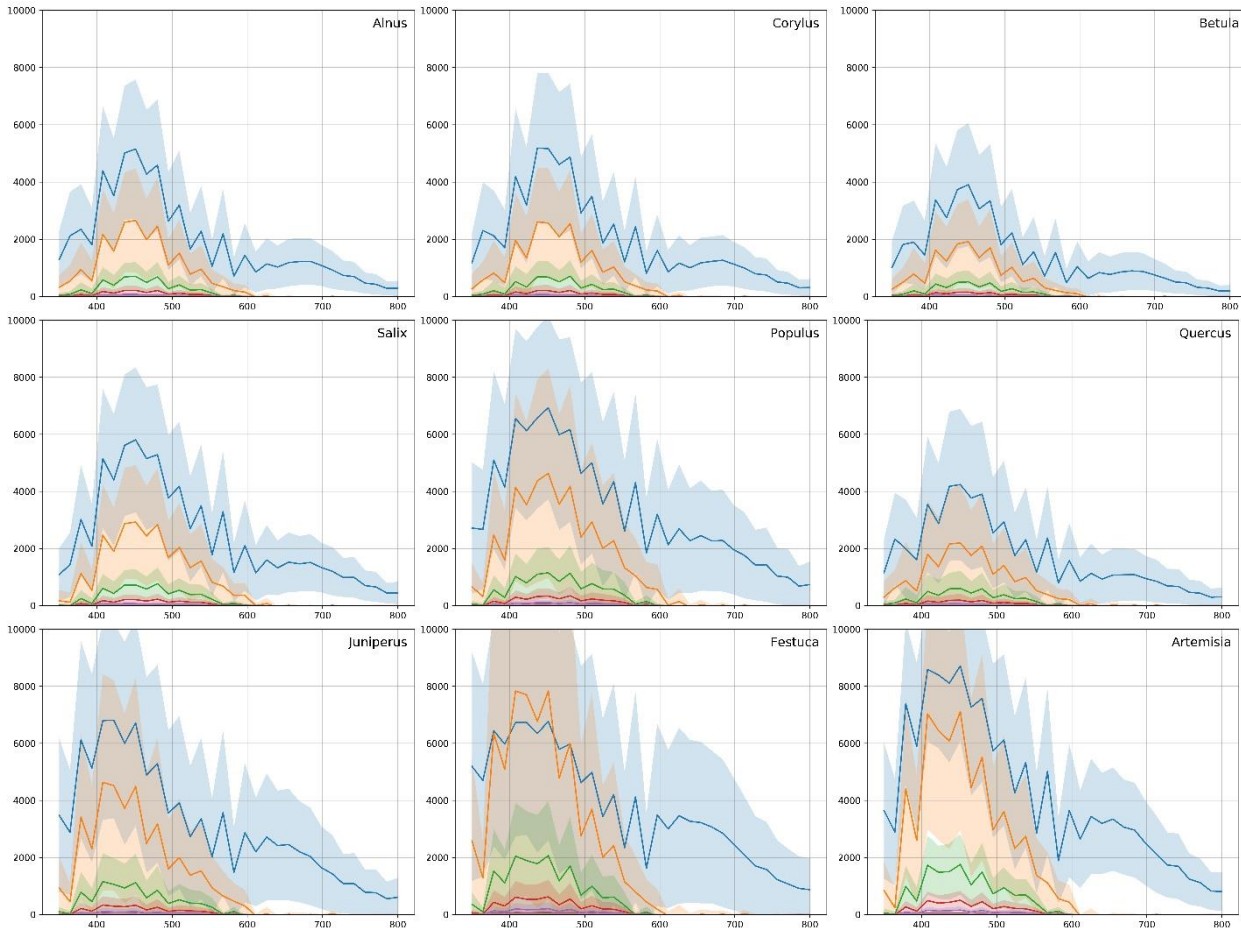

**Figure 8.** Comparison of fluorescence indicators of the tested pollen. The blue line represents the first acquisition. All other lines are delayed acquisition by step of 500 ns from the last. Shadows show the standard deviation ranges for each acquisition. In the figures, the x-axis represents the wavelength, nm; the y axis shows the amplitude, NA






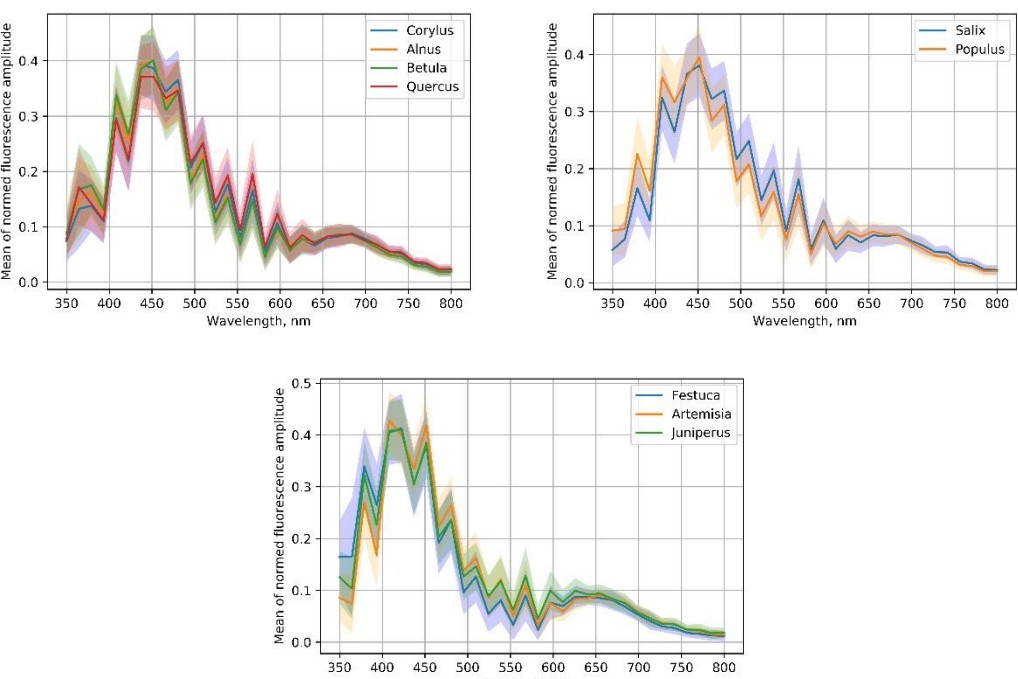

**Figure 9.** Groups with similar fluorescence spectra


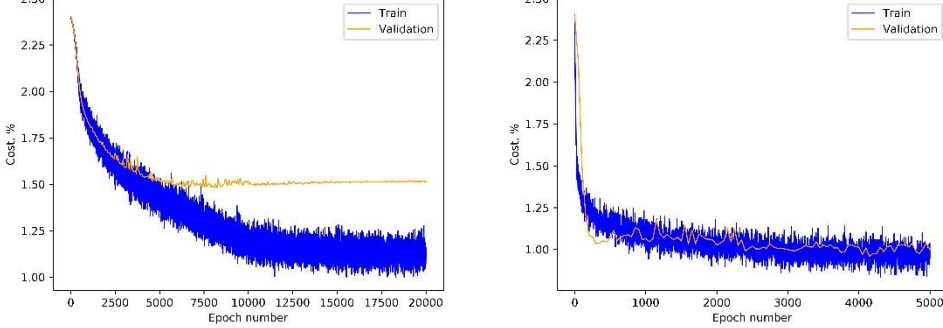

**Figure 10.** Siauliai ANN multi-species cost function for scatter (left) and fluorescence-(right) based recognition as a function of the training epoch.






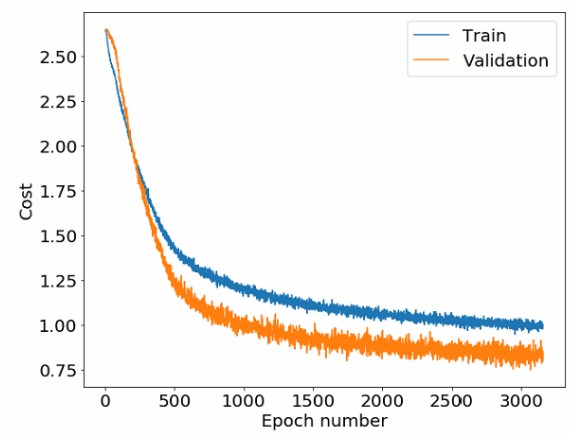

**Figure 11.** Novi Sad ANN overall cost as a function of the training epoch. The evaluation error is lower than the training error due to dropout (0.5) in each convolutional and fully connected layer, not used in the validation round.




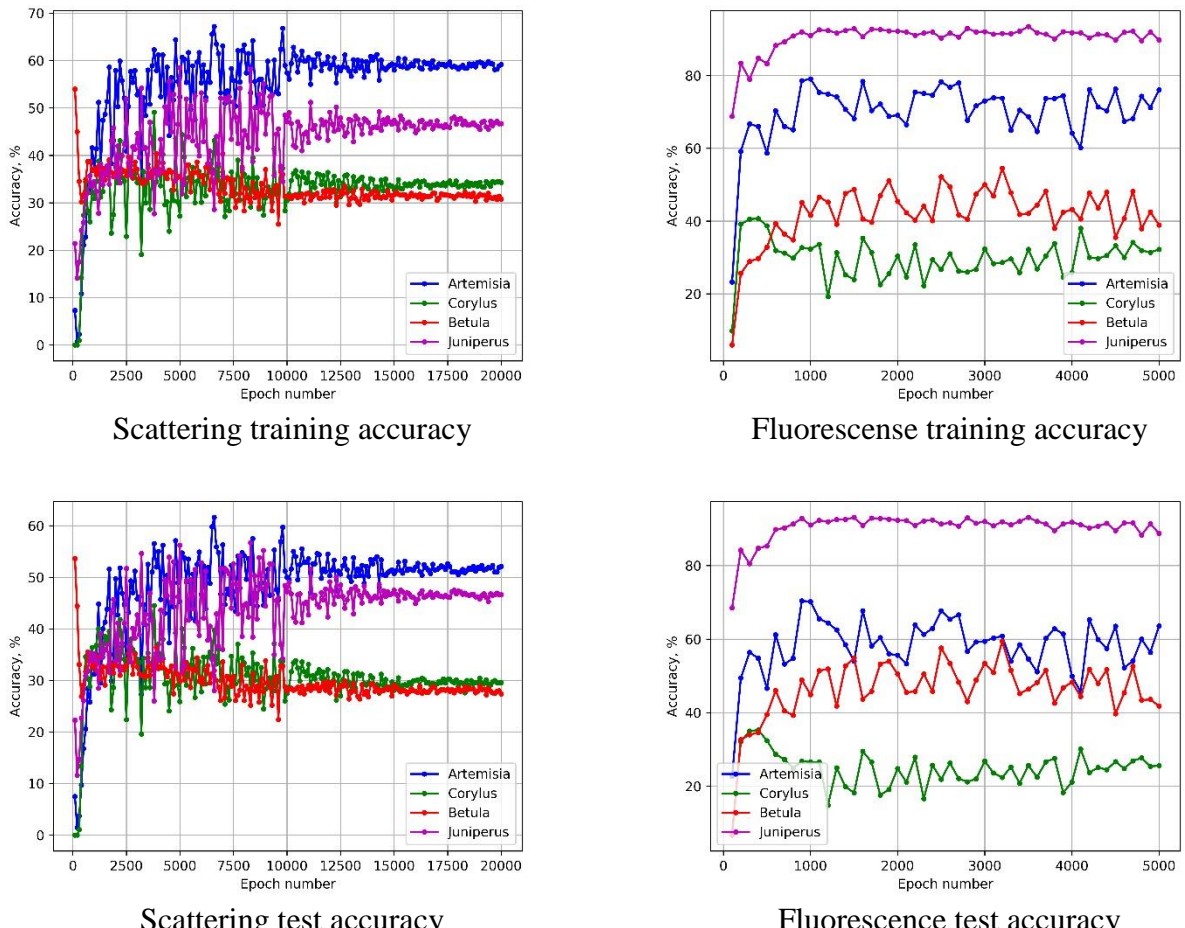

**Figure 12.** Performance of the Siauliai ANN for the test subset taken from different days than the calibration subset. Unit: %.





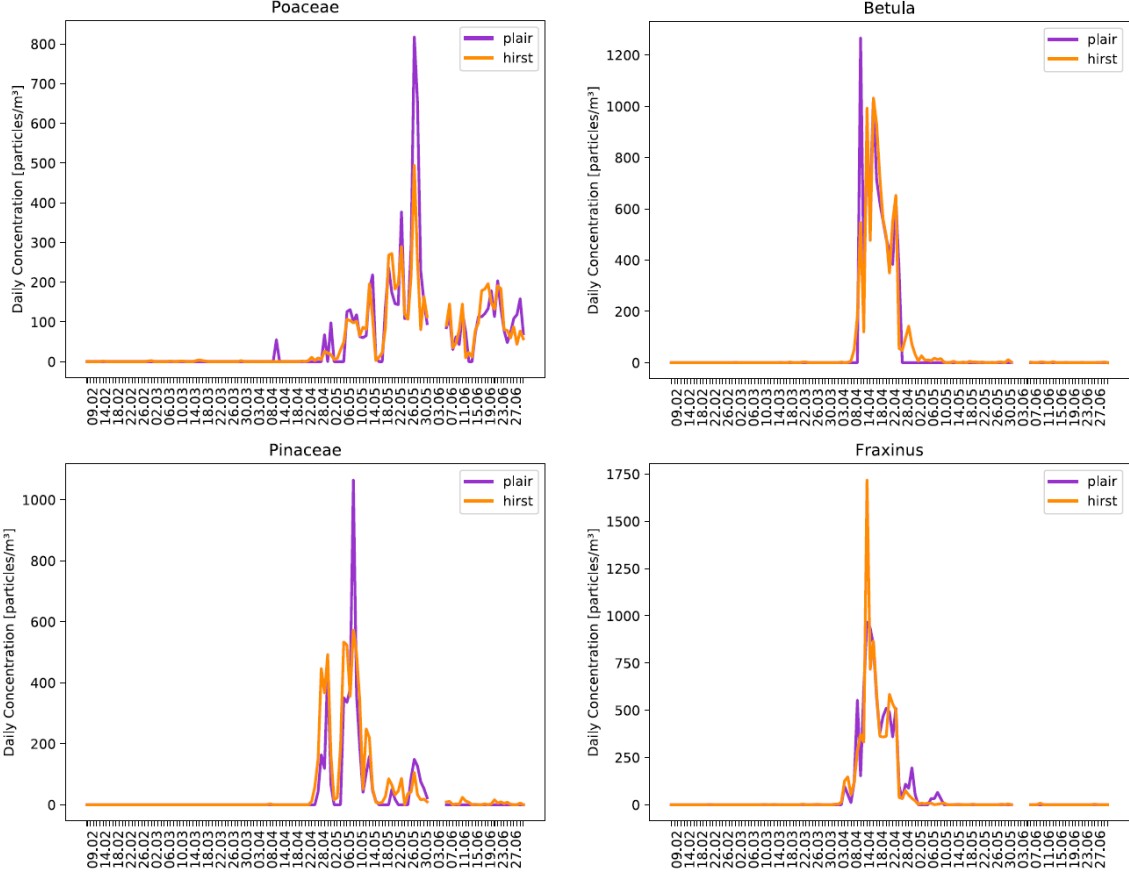

**Figure 13.** Comparison between automatic (Plair Rapid-E) and manual (Hirst-type) pollen counts for Betula, Fraxinus, Pinaceae, Poaceae.