# Peer review of "Automatic pollen recognition with the Rapid-E particle counter: the first-level procedure, experience and next steps"

_Atmospheric Measurement Techniques, 2018_

## Referee Comment (RC1) · Anonymous Referee #2 · 16 Feb 2019

The paper described an elaborate analysis of three different neural networks to classify pollen from scattering and fluorescence (and life time) imaging from three different centers in Europe, as a feasibility study. General Comments: 1. With three different research centers, three different training/test dataset, three different ANNs and three different experimental set-ups, it becomes unrealistic to make any meaningful comparisons. As a result, I'm afraid that the paper reads as if it were three separate studies intertwined, without any benefit of joining them into one. 2. The paper is quite lengthy and can be condensed considerably, improving the readability of the paper and preventing reader fatigue. 3. How do the results compare to the Poleno method that integrates both image recognition and laser fluorescence? 4. The timing of the flu-

orescence imaging is apparently such that it captures only a very limited amount of particles detected by the 400nm laser. As presented in Table 1, the recall rate is not only low in general; the percentage of analyzed particles was varying considerably among plant groups, ranging from 51% (for Quercus) to 87% (for Salix alba). Therefore, this raises the concern whether the first step already introduces a bias towards certain pollen types. 5. Experiments were carried out with fresh samples of collected pollen grains of separate plants and dried. Could this have skewed the results, as this procedure may decrease the amount of damaged pollen and aggregates as compared to daily practice? 6. What was the accuracy of the particle morphology step for pre-classifying pollen vs non-pollen? Is there any selection bias just based on particle size and therefore pollen type? 7. The reason for choosing three different designs of neural networks is not clear to me. Especially the choices in designing the first network from Siauliai seem to be a bit 'ad hoc'. In the network for analyzing the scattering images, it is surprising that three fully-connected layers were needed and that batch normal-ization was combined with dropout. In the network for fluorescence, it is not clear why no 1D convolution layers were used. Specific Comments: Introduction Generally, the introduction starts off quite clear, but at the end it needs more structure to clarify the goals of the paper. 8. line 52: "Hirst-type pollen traps" needs a bit more introduction on its methodology, before discussing its limitations. 9. line 72: Is there any literature on the evaluation of the Poleno device, that needs referencing? 10. line 77: the goal to evaluate the Rapid-E is described in too general terms. What aspect will be evaluated: sensitivity & specificity (and compared to which gold standard?), reproducibility of the system, processing speed or general applicability (Is this why three different centers participated?)? 11. line 78: what are "the Rapid-E data"? 12. line 81-85: Apparently, the different centers had different tasks in this project, but it is not clear which. As I understand it from the text, the system was assessed only in Siauliai, but then it is not clear how Novi Sad and Payerne determined their best classification. 13. line 82: Why did you compare the best classifications from the three centers, because they have different training or different procedures or pollen population? And what is the defini-

tion of 'the best classification'? 14. line 84: What was the hypothesis that led you to compare the Swiss classifier results to the Hirst data? And what is 'the Hirst' data? 15. line 96: which range do you refer to? 16. line 103-104: Do I understand correctly, that this follow-up analysis is then comparable to the Hirst-type of analysis? 17. line 109-110: What is the conclusion of this sentence? Is the 0.5-100um range not being used because of hardware life expectancy, or just because it does not apply to pollen? In the latter case, it is only logical to use the Pollen-mode. 18. line 111: Before entering the experiments section I would first expect a section on the ANN methods that do the classification, because these are evaluated in the experiments. 19. line 111: Chapter 2.2.The scheme of the experiment. Here I would expect first an overview of the different experiments that have been conducted, and especially the rationale behind them. As it is now, we dive into the details of the different experiments performed at the three sites, without a clue of the bigger picture. 20. line 113: What was the goal of this experiment (To test the accuracy)and what was the gold standard (separate purified samples of a particular plants)? Why was training needed in the first place (apparently the system does not come with a trained classifier for a specific area)? 21. line 117: "the pollen recognition algorithm" is not introduced yet. This is due to comment 14. 22. line 125: "repeated twice", so in total 3 times? 23. line 144: What does "practically identical" mean? So, the same classifier was trained (again) and the experiment repeated twice? 24. lines 150-156: It is not clear what the rationale is behind adjusting the test (and training?) set; is it to adapt to the local pollen population in Serbia or was it to test other hypotheses? 25. line 158: "For practical reasons" is a bit vague. 26. line 160: What were the threshold-based criteria? To what parameter was the threshold applied? 27. line 234: Why was a threshold of 10um used instead of 5? 28. lines 258: "Results obtained in Siauliai". Shouldn't there then be a chapter on the results of the other two sites? 29. lines 259-265: These research question should be presented in the introduction, not in the results section. 30. line 270: Why have the spectra been normalized, isn't the amplitude a characteristic in itself? 31. line 270-274: It is not clear what is tested statistically: difference between genera or within genera? The Student

test is not appropriate for testing data that is highly inter-correlated, like a spectrum. 32. line 72: "The uncertainties [. . .] were a fraction of a %.." This sentence is not clear to me: you mean less than 1%, but that is not true. 33. line 273: "statistically significant difference", what was the differences and what was the p-value? 34. line 323: Which of the 8 test data tests have been used here, what is n? (idem for section 3.2.2.) 35. line 372: "very similar way". I don't think you can say that since both the training and test data were different, and the more classes are included the more difficult the classification by an ANN will be. Technical Corrections: 36. line 72: typo, Swissens -> Swisens 37. line 89: typo, "analyses" -> "it analyses" 38. line 126: typo, "with" ->"with a" 39. line 128: typo, "fit" -> "fitted" 40. line 136: typo's, "compare" -> "compared" and "used new"-> "used a new" 41. line 202: typo: Rapid-e ->Rapid-E 42. line 205: typo, "founding" -> "finding" 43. line 150: style, "same or similar pollen morphotypes" -> "an adjusted set of pollen morphotypes" 44. line 217: NLL (negative log-likelihood) in full 45. line 810: Figure 1, the font size in this figure are quite small and difficult to read in a printed version. 46. line 820: The results of the two ANNs in Figure 3 and 4 were simply summed, whereas the other centers used concatenation. Then it is more consistent to show Figures 3 and 4 in one figure, and connect the two networks with a summation component. Please use the same conventions as in figures 5 and 6; the network in Figure 3 can be shortened by using Convolution Blocks, like in Figure 5. 47. line 825: Figure 6, for consistency with Figure 5, the lifetime and spectrometer sub-network need to be swapped. 48. line 835: Figure 8, font size on the x- and y-axis is too small to read in printed version. 49. line 830: y-axis in Figure 7, normed ->normalized 50. line 845: y-axis in Figure 9, normed ->normalized

---

## Referee Comment (RC2) · Anonymous Referee #3 · 19 Feb 2019

The study outlines the development of three artificial neural networks (ANN) at three different sites around Europe for use with a novel real-time fluorescence spectrometer instrument Rapid E from Plair. The need for such work is timely with allergy associated with pollen noted throughout the world and traditional methods sorely lack the time resolution to truly help those effected. The study does generate discussion on what the data analysis procedures should be for such instrumentation however it struggles to gel the three separate testing campaigns together well. Little in the way of true comparison is made. I believe the paper is useful, however the authors should be address the comments below

[Figure]

General comments (1) adding a column indicating the percentage of pollen that were deemed above the fluorescence threshold would be of help in table 1. Was this consistent from sample to sample? Ie Was approximately the same fraction of each pollen type deem fluorescent in each calibration test? (2) how reflective of pollen in the atmosphere would the test pollen be? The collection method used in payerne seems to significantly different than that seen in the other sites. (3) Was the compressed air, zero grade air from a cylinder or from a compressor for the Siauliai tests? (4) the instrument was run in pollen mode between 5-100 micron. Can this be changed? The majority of pollen is far larger than 5 micron. What is the expected lifetime of the instrument? Increasing the lower size threshold would likely extend it. Were the authors interested in fractionated pollen also? (5) L189: Do the authors have any idea how many pollen particles are not classified due to partly or fully missed the particle if the deep-UV laser fired at a wrong moment of time?

(6) L196: Is saturation of the fluorescence spectra still a possibility? Were some pollen more likely to cause this than others? (7) why was fluorescence lifetime not used in the Siauliai data analysis (8) how was the threshold of the particle fluorescence intensity level (> 1500 units) determined? This would be interesting for the reader and is generally discussed for other instruments? What was undertaken at the other sites? (9) for the Swiss data analysis, why was the optical size corresponding to 10 micrometers estimated? Was it simply due to the practicality of not having 10 micron PSLs? (10) The beginning of the results section has both general and site specific research questions (Lithuania). This seems out of place (consider moving to intro). (11) Consider cutting section 3.2.1. Recognition using scattering images only to a sentence and removing table 2 or moving it to a supplemental section. It does not add to the results and is far and away the weakest procedure. (12) Should there be results sections for the other two sites as well? Or is the results section an amalgam of the other sites also. Currently it reads like the results originate only from Siauliai (13) A very interesting observation that Festuca pollen was seen with the signal amplitude growing during the first 500 ns (Figure 8) do the author have any suggestion as to why

this is so? (14) A line on chemical interactions and degradation should be added L 405

(15) A brief mention of the work undertaken by the WIBS instrument should be discussed in 4.3. Comparison with other studies on pollen recognition (16) authors have suggested lifetime could be utilized to discriminate between pollen I feel this should be discussed, for example "O'Connor et al Using spectral analysis and fluorescence lifetimes to discriminate between grass and tree pollen for aerobiological applications Anal. Methods, 2014,6, 1633-1639" (17) What are the R2 values between the Rapid E and the Hirst? Is the Hirst a true reflection of what is in the atmosphere? (18) Could large fungal spores or clumps of fungal spores act as an interferant in the Rapid E for its current task? (19) Did only the Swiss site compare the Rapid E to a Hirst type trap? If yes why? If not, why are the results talked not about? This would be a good way to evaluate the ANN at each sampling site.

(20) Does the Rapid E come with any classifier? Or is it incumbent on the purchaser to develop their own? If the creator has an algorithm 5-10% better than seen here why is this not part of the commercial instrument?

Specific comments L19 specialized rather specialization L65 "it has become a necessity to develop new methods enabling the information on airborne pollen to become available in real-time" L66 were related L87 experiments L128 fitted rather than fit L136 a new bottle L137 "previously sampled" rather than "blown" L142 clarify for the reader what you mean by busy slides L212 which aimed L312 a challenging task

---

## Author Response (AR1)

Authors response to reviewer's comments.

First of all, we would like to thank the reviewers for their detailed comments to the paper. We used their comments and criticism to improve the manuscript. Below, we provide point-by-point response to the issues raised by the reviewers.

5

**Reviewer 2**

1. With three different research centers, three different training/test dataset, three different ANNs and three different experimental set-ups, it becomes unrealistic to make any meaningful comparisons. As a result, I'm afraid that the paper reads as if it were three separate studies intertwined, without any benefit of joining them into one

10 of joining them into one.

We respectfully disagree. The studies were indeed following their own paths, primarily reflecting the differences in the three independent projects supporting the work of the teams. It is also important that there is no previous experience with Rapid-E and no "good practice" could therefore be followed – development of such practice is one of the purposes of this paper. The main added value was that the

- 15 different approaches resulted in similar (but not completely identical) skills of the pollen recognition. The differences in the skills were, where possible, traced to the different approaches, ANN construction and data pre-processing. As a result, all groups participating in the paper were able to assess efficiency of their approaches and look into alternative ways of analysis. Now, we would like to share the findings with others, who are just considering to enter the field of online pollen monitoring. We hope that the revised
- 20 paper is more clear in this regard.

2. The paper is quite lengthy and can be condensed considerably, improving the readability of the paper and preventing reader fatigue.

We have reviewed the paper structure and shortened it

25

3. How do the results compare to the Poleno method that integrates both image recognition and laser fluorescence?

There was no comparison with POLENO monitor, it was not the purpose of the paper to compare different devices. We were building the procedures for Rapid-E and analyse the "absolute" skills of recognition 30 that can be obtained with the Rapid-E technology

4. The timing of the fluorescence imaging is apparently such that it captures only a very limited amount of particles detected by the 400nm laser. As presented in Table 1, the recall rate is not only low in general; the percentage of analysed particles was varying considerably among plant groups, ranging from 51%

35 (for Quercus) to 87% (for Salix alba). Therefore, this raises the concern whether the first step already introduces a bias towards certain pollen types.

The recall rate for fluorescence is indeed one of the significant concerns. For the confusion tables, however, the recall rate was not important: the initial number of pollen noticed by the scattering laser is not used in the analysis, i.e. the algorithms only dealt with fluorescent particles. The point is explicitly commented in the revised paper as a note to Table 1.

40 commented in the revised paper as a note to Table 1.

5. Experiments were carried out with fresh samples of collected pollen grains of separate plants and dried. Could this have skewed the results, as this procedure may decrease the amount of damaged pollen and aggregates as compared to daily practice?

45 This seems to be improbable, in fact, shaking pollens off the inflorescences and storing them in a package may rather increase the number of aggregates and small-size debris. This is why we put attention to microscopic analysis confirming that the fraction of these additions is very small.

6. What was the accuracy of the particle morphology step for preclassifying pollen vs non-pollen? Is there 50 any selection bias just based on particle size and therefore pollen type?

- In fact, the pollen-nonpollen pre-classification (so-called "pollen mode" of Rapid-E) is solely based on a requirement of the particle being larger than 5 um of optical diameter. Therefore, introduction of bias at this step is very improbable.
- 55 7. The reason for choosing three different designs of neural networks is not clear to me. Especially the choices in designing the first network from Siauliai seem to be a bit 'ad hoc'. In the network for analyzing the scattering images, it is surprising that three fully-connected layers were needed and that batch normalization was combined with dropout. In the network for fluorescence, it is not clear why no 1D convolution layers were used.
- 60 As stated above, there is no guidance of optimal practices of ANN construction for Rapid-E. Therefore, all participating groups were building their own approaches and all have tried several options before arriving at the ones shown in the paper. Harmonization effort is certainly in need but was left for the future when the problem of compatibility of the devices themselves is solved. In the Siauliai case, various options have been tested for the neural network design. The three layers gave the best results. To avoid
- 65 overtrainnig, we used regularization layers that are currently often combined: dropout and batch normalization. The 1D convolutional layers are suitable when the entire signal is homogenous. Pollen fluorescence peaks at different wavelengths are different, so it seemed expedient to use a fully-connected layer.
- 70 Introduction Generally, the introduction starts off quite clear, but at the end it needs more structure to clarify the goals of the paper.The goals of the paper have been refined

The goals of the paper have been refined

8. line 52: "Hirst-type pollen traps" needs a bit more introduction on its methodology, before discussing 75 its limitations.

An outline of the Hirst device is added

9. line 72: Is there any literature on the evaluation of the Poleno device, that needs referencing? Not really. This is a very new device and no consistent evaluation has been published.

80

10. line 77: the goal to evaluate the Rapid-E is described in too general terms. What aspect will be evaluated: sensitivity & specificity (and compared to which gold standard?), reproducibility of the system, processing speed or general applicability (Is this why three different centers participated?)?

Clarification is added, goals have been refined, as well as the roles of each centre. As stated above, the stated are operating within completely independent projects, which initiated certain diversity in the approaches.

11. line 78: what are "the Rapid-E data"? Clarified

90

12. line 81-85: Apparently, the different centers had different tasks in this project, but it is not clear which. As I understand it from the text, the system was assessed only in Siauliai, but then it is not clear how Novi Sad and Payerne determined their best classification.

Clarification is added, goals have been refined, as well as the roles of each centre. As stated above, the 95 centres are operating within completely independent projects, which initiated certain diversity in the approaches.

13. line 82: Why did you compare the best classifications from the three centers, because they have different training or different procedures or pollen population? And what is the definition of 'the best100 classification'?

Clarification added

14. line 84: What was the hypothesis that led you to compare the Swiss classifier results to the Hirst data? And what is 'the Hirst' data?

105 The selection of the group was opportunistic: the Swiss team had the Rapid-E and Hirst devices working next to each other during full seasons in the previous years, so it was possible to put the time series next to each other. However, the paper does not have a goal of the Rapid-E – Hirst comparison. The provided time series were added to discuss the "false positive" identifications. We made it clearer in the revised version

110

15. line 96: which range do you refer to?

Clarified: changed to "this saturation level will not be reached in realistic ambient conditions"

16. line 103-104: Do I understand correctly, that this follow-up analysis is then comparable to the Hirst-115 type of analysis?

Well, to some extent. The geometry of the slide and flow is strongly different but the principle is the same: impaction on a sticky slide followed by the microscopic analysis.

17. line 109-110: What is the conclusion of this sentence? Is the 0.5-100um range not being used becauseof hardware life expectancy, or just because it does not apply to pollen? In the latter case, it is only logical to use the Pollen-mode.

In this section, we did not derive conclusions, only described the device. The pollen mode as the default one is brought up later when we describe the setup of the experiment.

125 18. line 111: Before entering the experiments section I would first expect a section on the ANN methods that do the classification, because these are evaluated in the experiments. The data analysis sections and the experiment description sections have been switched

19. line 111: Chapter 2.2. The scheme of the experiment. Here I would expect first an overview of the

130 different experiments that have been conducted, and especially the rationale behind them. As it is now, we dive into the details of the different experiments performed at the three sites, without a clue of the bigger picture.

The experimental section has been rearranged to provide the overall picture first and then specificity of each site.

135

20. line 113: What was the goal of this experiment (To test the accuracy) and what was the gold standard (separate purified samples of a particular plants)? Why was training needed in the first place (apparently the system does not come with a trained classifier for a specific area)?

Indeed, the system does not come with any classifier to any of the machines, plus the machines proved to 140 be substantially different.

21. line 117: "the pollen recognition algorithm" is not introduced yet. This is due to comment 14. The sentence is reformulated

145 22. line 125: "repeated twice", so in total 3 times?

No, just twice. The word "repeated" is replaced with "performed"

23. line 144: What does "practically identical" mean? So, the same classifier was trained (again) and the experiment repeated twice?

150 No, just the practical side, i.e. the pollen exposure. Clarified.

24. lines 150-156: It is not clear what the rationale is behind adjusting the test (and training?) set; is it to adapt to the local pollen population in Serbia or was it to test other hypotheses? This is indeed just reflecting the pollen availability in Serbia.

**155**

25. line 158: "For practical reasons" is a bit vague.

The meaning clarified: "Low ambient concentration of coarse particles allowed a less laborious approach: pollen calibrations in Payerne ...."

160 26. line 160: What were the threshold-based criteria? To what parameter was the threshold applied? Clarified

27. line 234: Why was a threshold of 10um used instead of 5?

Partly, for the historical reasons, following the work of Crouzy et al. For the tested pollens it is the same:165 they all are much larger. But the tighter threshold was effective in eliminating the ambient particles, which, albeit in low numbers, were still reaching the device.

28. lines 258: "Results obtained in Siauliai". Shouldn't there then be a chapter on the results of the other two sites?

170 The results from all three sites are collected together

29. lines 259-265: These research question should be presented in the introduction, not in the results section.

Moved to introduction

**175**

30. line 270: Why have the spectra been normalized, isn't the amplitude a characteristic in itself? Each spectrum was normalised separately prior to any further analysis to eliminate the effect of imperfect hit of the pollen by the UV laser impulse. Normalization allows to reduce variation of fluorescence amplitude for one pollen taxon at fixed wavelength.

**180**

31. line 270-274: It is not clear what is tested statistically: difference between genera or within genera? The Student test is not appropriate for testing data that is highly inter-correlated, like a spectrum. In the shortened paper, we removed the Student test remark as a weakly-connected part of the analysis.

185 32. line 272: "The uncertainties [...] were a fraction of a %.." This sentence is not clear to me: you mean less than 1%, but that is not true.

Corrected to "percentage".

33. line 273: "statistically significant difference", what was the differences and what was the p-value? Updated to p < 0.001.

**190**

34. line 323: Which of the 8 test data tests have been used here, what is n? (idem for section 3.2.2.) In Siauliai experiment with each genus consisted of up to 8 sample tests (approximately 5 mg of pollen/test). n of registered data by Rapid-E and analysed in research is indicated in Table 1 in column "Fluorescent particles". They were all subjected to the recognition analysis, together.

**195**

35. line 372: "very similar way". I don't think you can say that since both the training and test data were different, and the more classes are included the more difficult the classification by an ANN will be. The sentence is changed to "showed similar recognition skills"

**200 Technical Corrections:**

36. line 372: typo, Swissens ->Swisens 37. line 89: typo, "analyses" -> "it analyses"

38. line 126: typo, "with" ->"with a"

- 39. line 128: typo, "fit" -> "fitted"
- 40. line 136: typo's, "compare" -> "compared" and "used new"-> "used a new"
- 205 41. line 202: typo: Rapid-e ->Rapid-E
  - 42. line 205: typo, "founding" -> "finding"
  - 43. line 150: style, "same or similar pollen morphotypes" -> "an adjusted set of pollen morphotypes"
  - 44. line 217: NLL (negative log-likelihood) in full

45. line 810: Figure 1, the font size in this figure are quite small and difficult to read in a printed version. 210 Thanks you! All corrected

46. line 820: The results of the two ANNs in Figure 3 and 4 were simply summed, whereas the other centers used concatenation. Then it is more consistent to show Figures 3 and 4 in one figure, and connect the two networks with a summation component. Please use the same conventions as in figures 5 and 6;

215 the network in Figure 3 can be shortened by using Convolution Blocks, like in Figure 5. Corrected

47. line 825: Figure 6, for consistency with Figure 5, the lifetime and spectrometer sub-network need to be swapped.

220 48. line 835: Figure 8, font size on the x- and y-axis is too small to read in printed version.

49. line 830: y-axis in Figure 7, normed ->normalized 50. line 845: y-axis in Figure 9, normed ->normalized

Thank you! All corrected

**225**

**Reviewer 3**

General comments

(1) adding a column indicating the percentage of pollen that were deemed above the fluorescence threshold would be of help in table 1. Was this consistent from sample to sample? Ie Was approximately230 the same fraction of each pollen type deem fluorescent in each calibration test?

The column has been added, together with the comment that the fractions were not exactly the same.

(2) how reflective of pollen in the atmosphere would the test pollen be? The collection method used in Payerne seems to significantly different than that seen in the other sites.

- 235 The basic idea was to use as fresh pollen as possible but after drying it somewhat. This is generally consistent with what is happening in the environment because pollen grains tend to loose water during the first minutes of the atmospheric transport providing that the weather is good and relative humidity is comparatively low also the conditions facilitating the pollen release.
- 240 (3) Was the compressed air, zero grade air from a cylinder or from a compressor for the Siauliai tests? Zero grade air from the cylinder.

(4) the instrument was run in pollen mode between 5-100 micron. Can this be changed? The majority of pollen is far larger than 5 micron. What is the expected lifetime of the instrument? Increasing the lower245 size threshold would likely extend it. Were the authors interested in fractionated pollen also?

The lower bound of 5 micrometres is the factory setting, hardcoded into hardware. The lifetime of the instrument is yet-unknown but estimates circle around 3-4 years depending on the level of pollution, amount of calibration exercises, etc. None of our instruments reached its end till today.

- 250 (5) L189: Do the authors have any idea how many pollen particles are not classified due to partly or fully missed the particle if the deep-UV laser fired at a wrong moment of time? In the table 1, the "fluorescent" particles compare to "total particles" is the fraction of the sufficiently good hits.
- 255 (6) L196: Is saturation of the fluorescence spectra still a possibility? Were some pollen more likely to cause this than others?

The devices behaved differently: the saturation was more a problem in Siauliai than in other labs. But after exclusion of the first spectrum, the impact of this problem became small.

260 (7) why was fluorescence lifetime not used in the Siauliai data analysis

It has been noticed that the lifetime signal was often saturated in-between the rise and fall sections when the value remains constant over some time.

(8) how was the threshold of the particle fluorescence intensity level (> 1500 units) determined? This would be interesting for the reader and is generally discussed for other instruments? What was undertaken

265 at the other sites?

This threshold is an empirical parameter and it had to vary between the devices. Somehow, the strength of the signal was substantially different between the labs. This was one of the reasons for normalization of the spectra. We added a clarifying sentence in the revised paper.

270 (9) for the Swiss data analysis, why was the optical size corresponding to 10 micrometers estimated? Was it simply due to the practicality of not having 10 micron PSLs?

Partly, for the historical reasons, following the work of Crouzy et al. For the tested pollens it is the same: they all are much larger. But the tighter threshold was effective in eliminating the ambient particles, which, despite in low numbers, were still reaching the device.

275

(10) The beginning of the results section has both general and site specific research questions (Lithuania). This seems out of place (consider moving to intro).

The results from all three sites are collected together

280 (11) Consider cutting section 3.2.1. Recognition using scattering images only to a sentence and removing table 2 or moving it to a supplemental section. It does not add to the results and is far and away the weakest procedure.

We have moved the details of the separate recognitions with scattering and fluorescence only into an Annex but still consider this analysis useful as it shows the relative importance of each information 285 channel. Therefore, a few sentences comparing these channels are retained in the revised paper.

(12) Should there be results sections for the other two sites as well? Or is the results section an amalgam of the other sites also. Currently it reads like the results originate only from Siauliai

The results from all three sites are collected together

(13) A very interesting observation that Festuca pollen was seen with the signal amplitude growing during the first 500 ns (Figure 8) do the author have any suggestion as to why this is so? Unfortunately, no, this is just the empirical fact.

295 (14) A line on chemical interactions and degradation should be added L 405 A sentence is added

(15) A brief mention of the work undertaken by the WIBS instrument should be discussed in 4.3. Comparison with other studies on pollen recognition

300 A short discussion is added. We added information to Introduction and 4.3 section.

(16) authors have suggested lifetime could be utilized to discriminate between pollen I feel this should be discussed, for example "O'Connor et al Using spectral analysis and fluorescence lifetimes to discriminate between grass and tree pollen for aerobiological applications Anal. Methods, 2014,6, 1633-1639"

305 Thank you for the reference! We have added the reference and corresponding discussion.

(17) What are the R2 values between the Rapid E and the Hirst? Is the Hirst a true reflection of what is in the atmosphere?

The R2 values are provided. Hirst is certainly not the true reflection but the best we have for last 60 years

310

(18) Could large fungal spores or clumps of fungal spores act as an interferant in the Rapid E for its current task?

No, they will not. During next steps of calibrations, not shown in this paper, we included spores and the first impression is that the difference is substantial, first of all, in fluorescence spectra, which do not

315 depend on agglomeration.

(19) Did only the Swiss site compare the Rapid E to a Hirst type trap? If yes why? If not, why are the results talked not about? This would be a good way to evaluate the ANN at each sampling site.

Not only but the Payerne group has the longest experience with the Plair device (albeit with its previous 320 version). Therefore, they faced the problem of the false-positives, which is discussed in this section.

(20) Does the Rapid E come with any classifier? Or is it incumbent on the purchaser to develop their own? If the creator has an algorithm 5-10% better than seen here why is this not part of the commercial instrument?

325 In theory, it can have it. However, none of our devices were equipped with it. In all cases, the algorithm is closed and requires calibration to adapt to the specific machine. All machines are provided as "experimental devices", with reduced warranty period and some features disabled. However, the conclusions of the paper imply that the details of such algorithm are not too relevant: pre-processing and pre-filtering the input datasets may have stronger influence on the recognition quality than setup of the 330 ANN.

Specific comments

L19 specialized rather specialization

335 L65 "it has become a necessity to develop new methods enabling the information on airborne pollen to become available in real-time"

L66 were related

L87 experiments

L128 fitted rather than fit

340 L136 a new bottle

L137 "previously sampled" rather than "blown"

L142 clarify for the reader what you mean by busy slides

L212 which aimed

L312 a challenging task

345 Thank you! All corrected

On behalf of authors Ingrida Šaulienė

**Automatic pollen recognition with the Rapid-E particle counter: the first-level procedure, 355 experience and next steps**

Ingrida Šaulienė1, Laura Šukienė1, Gintautas Daunys1, Gediminas Valiulis1, Lukas Vaitkevičius1, Predrag Matavulj2, Sanja Brdar2, Marko Panic2, Branko Sikoparija2, Bernard Clot3, Benoît Crouzy3 Mikhail Sofiev1,4

[revised manuscript text omitted]

- 445 recognition algorithms have been constructed independently were carried out in each of the three-center.s This organization of the study allowed accounting for until an algorithm could identify the selected pollen. variability of the actual All devices are different and the same algorithm cannot be successfully used due to the technical characteristics of the individual devices and an absence of "good practice" for such type of measurements. The best classification results is then compared across the centres formed the basis of
- 450 the Result section of this paperwith the best first level classification of same or similar pollen morphotypes made independently in Novi Sad and Payerne. Finally, outputs of the MeteoSwiss classifier in a monitoring setup are presented and compared with airborne pollen data collected with by using the Hirst-type pollen trap data. The provided time series were used added to discuss the "false positive" identifications important for the operational context.

**455 2 Methods**

**2.1. Description of the measurement instrument**

The new Rapid-E instrument designed and produced by Plair S.A is the successor of the first-generation particle analyser PA-300 used by Crouzy et al. (2016). It is a particle counter, i.e. it analyses all particles coming to its inlet one-by-one. Operation of the instrument is based on two physical principles: scattering 460 of near-UV laser beam and deep-UV laser-induced fluorescence (Kiselev et al., 2011; 2013). Multi-angle scattering is used for determination of the particle's morphology, such as size and shape. The fluorescent light is analysed for its spectrum and lifetime. The instrument constantly takes in the ambient air through the air inlet on the top of its panel. Sample air flow is up to 2.8 litres per minute with the counting rate of up to 4500 particle detections per minute, i.e. the theoretical saturation level is 1.6 106 particles m-3. Since
465 according to the device provider the smallest observable particle is 0.5µm in diameter, this range is sufficient for practically all saturation level will not be reached in realistic ambient conditions.

The sampled air enters the nozzle, which creates a laminar flow in the measurement zone. Particles interact with 400 nm laser light source and the scattered light is captured by twenty-four time-resolving detectors distributed at different angles. The information on chemical properties of the particles is

470 obtained by a powerful deep-UV laser (320 nm) source that induces fluorescence. Its spectrum (32 measuring channels within spectral range of 350-800 nm, 8 sequential acquisitions with 500 ns retention) and lifetime (4 particular bands) are recorded and used for the particle identification (Figure 1). The threshold of the particle fluorescence intensity level (> 1500 units) was empirically determined as a cut-off level for sufficiently recorded pollen grains. Tfor this research and the spectra wereas subsequently

475 normalized due-to eliminate the difference in the signal strength of the signal was substantially different between the measurement-instruments.

Rapid-E has an embedded mechanism for collecting the particles, which passed through the registration chamber, onto sticky slides for the follow-up microscopic analysis.

- 480 The device has several modes of operations. Since the deep-UV laser has a limited resource, the 400 nm scattering image is used for prior estimation of the particle morphology and deciding if it can be pollen. In the Pollen mode, the device ignites the deep-UV laser only for 5-100 micrometer particle size range (used in for-this studyresearch). Solely based on a requirement of the particle being larger than 5 um of optical diameter the Pollen mode was used in pollen-nonpollen pre-classification. 
[revised manuscript text omitted]
 study-the baseline (groundwork)-calibration experiment of how Rapid-E can identify different species of pollen genera or pollen within the genus-was implemented in Siauliai, followed -(Lithuania). The by the description of specifics of the setups in unique results of Novi Sad (Serbia) and Payerne (Switzerland) experiments were integrated into the accuracy evaluation in different geographic territories. The Most-importantly, pollen we used
- 620 in the study was only typical pollen characteristic for each particular location was used. For eComparison of the results were based on purposes, the selected pollen types belonging to the same plant families that are found in all three locations.

**2.3.1 Siauliai**

625 The experiment in Siauliai was carried out with 14 pollen morphotypes, the tested amounts of which are given in Table 1. Three genera (*Salix, Acer, Pinus*) were represented by two plant types. All 14 plants are naturally widespread in Lithuania and their airborne pollen is abundantly recorded annually (Šaulienė et al., 2016). These particles were provided to the device one set after another splitting the recordings to

training and test subsets. After that, the pollen recognition algorithm was calibrated using training subsets 630 and challenged with pollen from the test subsets with no stratification or a priori other information

- Pollen was taken from the plant inflorescences collected during the vegetation period in April-August of 2018 during the days with intense pollen release. The collected material was put in air-permeable paper bags and dried at a temperature of 40°C until the maximum release of pollen from the inflorescences.
- 635 Vibratory Sieve Shaker ANALYSETTE 3 PRO was used for gentle shaking the pollen grains out of the inflorescences. The extracted pollen was stored in Petri dishes at +4°C. Each experiment was repeated performed twice and consisted of up to eight 8 sample tests, each using approximately 5 mg of pollen per sample test. The number n-of grains registered in the scattering signal data by Rapid-E and analysed in research-
[revised manuscript text omitted]
     | <del>Corylus</del> | Alnus         | <del>Betula</del> | Saliv         | Populus       | Acer          | Junip cruss   | <del>Quereus</del> | Dinus         |    |
|--------------------|---------------|---------------|--------------------|---------------|-------------------|---------------|---------------|---------------|---------------|--------------------|---------------|----|
| Festuca            | <del>52</del> | 1             | 1                  | 7             | 1                 | 3             | 3             | 5             | 5             | 9                  | 19     |    |
| Artemisia          | 7             | <del>58</del> | 1                  | 1             | 4                 | 13     | 3             | 4             | 9             | 3                  | 2             | ]  |
| <del>Corylus</del> | 3             | 1             | 38                 | <del>20</del> | <del>20</del>     | 1             | 1             | 5             | 1             | 9                  | 1             | ]  |
| Alnus              | 4             | 3             | $\frac{21}{21}$    | <del>29</del> | 19         | 3             | 1             | 6             | 1             | 11                 | 7      | 1. |
| <del>Betula</del>  | 4             | 4             | 19          | <del>15</del> | <del>37</del>     | 3             | 1             | 5             | 1             | 9                  | 2             |    |
| <del>Salix</del>   | 5             | 8             | 1                  | 0             | 5                 | <del>51</del> | <del>13</del> | 3             | 6             | 7                  | 1             | 1  |
| Populus            | <del>16</del> | 4             | 0                  | 1             | 1                 | <del>16</del> | <del>28</del> | 4             | <del>16</del> | 8                  | 6             | 1  |
| Acer               | 8             | 5             | 6                  | 7             | 4                 | 4             | 5             | <del>29</del> | 4             | 23                 | 5             | 1  |
| Juniperus          | <del>16</del> | 19     | 0                  | 0             | 0                 | 4             | <del>10</del> | 2             | 36            | 3                  | <del>10</del> | ]  |
| <del>Quercus</del> | 7             | 2             | 10                 | 6             | 3                 | 3             | 3             | 14            | 2             | <del>48</del>      | 2             | 1  |
| Pinus              | 13            | 2             | 0                  | 1             | 1                 | 1             | 1             | 1             | 3             | 1                  | 76            | 1  |

Predicted label

**Table 3.** Confusion table for pollen taxa identification using fluorescence spectrum.**

| Plant genus        | Festuca       | Artemisia     | <del>Corylus</del> | Almus     | <del>Betula</del> | Saliv         | Populus       | Acer          | Juniperus     | <del>Quereus</del> | Pinus  |     |
|--------------------|---------------|---------------|--------------------|-----------|-------------------|---------------|---------------|---------------|---------------|--------------------|---------------|-----|
| Festuca     | <del>84</del> | 7             | 0                  | 0         | 0                 | θ             | 1             | 5             | 3             | 0                  | чħ            |     |
| Artemisia          | 7             | <del>80</del> | 0                  | 0         | 0                 | 1             | 3             | 6             | 8             | 0                  | 0             |     |
| <del>Corylus</del> | 7             | 0             | <del>53</del>      | 15        | 7                 | 6             | 1             | 0             | 1             | 14                 | 1             |     |
| Alnus              | 1             | 0             | 14          | 45 | 19         | 3             | 4             | 1             | 1             | <del>10</del>      | Ð             | _   |
| Betula             | 3             | 1             | 7                  | 29        | 38                | 3             | 4             | 0             | 1             | 12                 | Ð             | ्रि |
| <del>Salix</del>   | 0             | 7             | 7                  | 7         | 3                 | 72            | 9             | 0             | 1             | 4                  | 0             | ٦ ا |
| Populus            | ₹             | ₽             | 3                  | ₹         | ₹                 | <del>17</del> | <del>53</del> | 4             | 4             | 3           | ŧ             |     |
| Acer               | €             |        | 1                  | 0         | 1                 | ₽             | ₹             | <del>84</del> | ₹             | 1                  | 0             |     |
| Juniperus          | 6             | 5             | 0                  | 0         | 1                 | 0             | 3             | 0             | <del>84</del> | 0                  | ŧ             |     |
| <del>Quercus</del> | Ŧ             | 0             | 12                 | 13        | 9          | 6             | Ŧ             | 0             | 1
1        | <del>55</del>      | ŧ             |     |
| <del>Pinus</del>   | ##            | ₽             | 0                  | ₽         | 1                 | 0             | 0             | ₽             | 1             | ŧ                  | <del>86</del> |     |

Multiclass accuracy 67 %

Predicted label

 Table 42. Confusion table obtained in Siauliai. based on two ANNs fused by summing scores of every pollen type

1375

|             |         |           |         | Accu  | iracy  | v: 73 | %       |      |           |         |       | _   |
|-------------|---------|-----------|---------|-------|--------|--------------|---------|------|-----------|---------|-------|-----|
| Plant genus | Festuca | Artemisia | Corylus | Alnus | Betula | Salix        | Populus | Acer | Juniperus | Quercus | Pinus |     |
| Festuca     | 88      | 1         | 0       | 0     | 0      | 0            | 2       | 5    | 2         | 0       | 2     |     |
| Artemisia   | 2       | 86        | 0       | 0     | 0      | 2            | 1       | 4    | 5         | 0       | 0     | ]   |
| Corylus     | 2       | 0         | 63      | 17    | 8      | 1            | 0       | 0    | 0         | 9       | 0     | ]   |
| Alnus       | 1       | 0         | 15      | 53    | 18     | 2            | 1       | 0    | 0         | 9       | 1     |     |
| Betula      | 3       | 1         | 9       | 30    | 47     | 1            | 1       | 0    | 1         | 6       | 1     | -   |
| Salix       | 1       | 1         | 2       | 1     | 2      | 78    | 10      | 0    | 1         | 4       | 0     |     |
| Populus     | 3       | 6         | 1       | 1     | 1      | 18           | 58      | 3    | 3         | 5       | 1     | ] _ |
| Acer        | 5       | 2         | 1       | 1     | 1      | 0            | 2       | 86   | 1         | 1       | 0     |     |
| Juniperus   | 4       | 4         | 0       | 0     | 0      | 0            | 3       | 1    | 87        | 0       | 1     | 1   |
| Quercus     | 2       | 0         | 9       | 10    | 5      | 4            | 1       | 0    | 0         | 69      | 0     | ]   |
| Pinus       | 7       | 0         | 0       | 0     | 0      | 0            | 0       | 0    | 1         | 0       | 91    |     |

Predicted label

**Table. 53. Confusion table obtained in Novi Sad. Accuracy 74% (obs different number of species)**

| Plant genus | Dactilis  | Cynodon | Corylus | Alnus | Betula | Salix | Fraxinus | Populus | Acer | Artemisia | Taxus | Quercus | Picea | Cedrus |     |
|-------------|-----------|---------|---------|-------|--------|-------|----------|---------|------|-----------|-------|---------|-------|--------|-----|
| Dactilis    | 78 | 3       | 0       | 3     | 0      | 2     | 0        | 4       | 5    | 0         | 1     | 0       | 0     | 0      |     |
| Cynodon     | 4         | 70      | 0       | 0     | 0      | 6     | 1        | 0       | 0    | 12        | 5     | 0       | 2     | 0      |     |
| Corylus     | 0         | 0       | 64      | 6     | 12     | 1     | 10       | 0       | 0    | 0         | 4     | 2       | 0     | 0      |     |
| Alnus       | 1         | 2       | 6       | 72    | 3      | 2     | 3        | 2       | 1    | 2         | 6     | 3       | 0     | 0      |     |
| Betula      | 1         | 0       | 25      | 5     | 51     | 3     | 3        | 0       | 0    | 1         | 1     | 10      | 1     | 0      |     |
| Salix       | 3         | 1       | 0       | 2     | 1      | 80    | 3        | 2       | 2    | 2         | 1     | 3       | 1     | 0      | el  |
| Fraxinus    | 0         | 0       | 7       | 1     | 3      | 2     | 79       | 1       | 0    | 0         | 4     | 3       | 0     | 0      | lab |
| Populus     | 5         | 3       | 1       | 4     | 0      | 4     | 3        | 71      | 1    | 1         | 3     | 4       | 1     | 0      | an. |
| Acer        | 8         | 1       | 0       | 2     | 1      | 4     | 0        | 0       | 73   | 0         | 1     | 9       | 1     | 0      | Tr  |
| Artemisia   | 1         | 5       | 0       | 2     | 1      | 4     | 0        | 1       | 0    | 84        | 0     | 1       | 1     | 0      |     |
| Taxus       | 0         | 3       | 0       | 2     | 0      | 0     | 0        | 0       | 0    | 0         | 93    | 1       | 1     | 0      |     |
| Quercus     | 1         | 0       | 4       | 5     | 8      | 9     | 2        | 1       | 1    | 1         | 1     | 63      | 4     | 0      |     |
| Picea       | 3         | 3       | 0       | 2     | 4      | 3     | 0        | 1       | 1    | 7         | 1     | 13      | 61    | 0      |     |
| Cedrus      | 0         | 0       | 0       | 0     | 0      | 1     | 0        | 1       | 0    | 1         | 0     | 0       | 1     | 95     |     |

Predicted label

**1380 **Table 6**4Confusion table obtained at MeteoSwiss, Payerne. Accuracy 80% (obs different number of species)**

| Plant genus | lnus | etula | arpinus | upressus | agus          | raxinus | orylus | inus          | oaceae        | axus   |       |
|-------------|------|-------|---------|----------|---------------|---------|--------|---------------|---------------|--------|-------|
| Alnus       | 27   | 27    | 1       | 0        | u
0 | 1       | 43     | d
0 | d
0 | L
1 |       |
| Betula      | 1    | 83    | 2       | 0        | 0             | 4       | 7      | 0             | 0             | 1      |       |
| Carpinus    | 0    | 13    | 74      | 0        | 0             | 2       | 3      | 0             | 6             | 1      | oel j |
| Cupressus   | 0    | 3     | 1       | 84       | 0             | 0       | 1      | 2             | 1             | 8      | lab   |
| Fagus       | 0    | 2     | 3       | 1        | 88            | 0       | 1      | 1             | 2             | 3      | rue   |
| Fraxinus    | 0    | 12    | 2       | 0        | 0             | 78      | 2      | 1             | 2             | 3      | Ē     |
| Corylus     | 4    | 8     | 0       | 0        | 0             | 0       | 87     | 0             | 0             | 0      |       |
| Pinus       | 0    | 0     | 0       | 0        | 0             | 0       | 0      | 98     | 0             | 2      |       |
| Poaceae     | 0    | 3     | 8       | 1        | 0             | 1       | 0      | 1             | 82            | 4      |       |
| Taxus       | 0    | 0     | 0       | 1        | 0             | 0       | 0      | 1             | 0             | 97     |       |
|             |      |       |         | Pred     | licted        | label   |        |               |               |        |       |